# Fair Classification with Efficient and Post-hoc Controllable Fairness-Accuracy Trade-off

**Maaya Sakata** [* 1 2] **Kazuto Fukuchi** [* 1 2]

## Abstract

Post-hoc controllability of fair machine learning models, the ability to control the trade-off between fairness and accuracy after training, is valuable for practical deployment. Existing post-processing methods provide such post-hoc controllability but often suffer from significant accuracy degradation, whereas in-processing methods achieve efficient trade-offs but require computationally expensive retraining for each change in trade-off ratio. To achieve both post-hoc controllability and efficient trade-offs, we propose a novel fair classification algorithm that learns effective feature representations to improve the trade-off efficiency of post-processing fair classifiers, by a gradient-based optimization approach. Experimental results on real-world datasets demonstrate that our method achieves trade-off efficiency comparable to, or even surpassing, in-processing methods, without requiring any retraining.

## 1. Introduction

While machine learning (ML)-based systems play a critical role in real-world decision making across various domains due to their strong predictive performance, these systems can suffer from inherent biases that lead to unfair outcomes. Indeed, many researchers have reported discriminatory outcomes of real-world ML-based systems against certain protected groups, such as those defined by gender, race, and age. For example, Amazon discontinued the development of an ML-based hiring system after it was found to exhibit gender bias (Dastin, 2018). Similar issues have been reported across various domains, including hiring (Fabris et al., 2025) and criminal justice (Angwin et al., 2016).

Mitigating these biases in ML systems is in high demand;

however, it must be achieved while carefully managing the trade-off between predictive accuracy and fairness. Accuracy and fairness are often in tension, as improvements in fairness may come at the cost of reduced predictive performance. In practice, high accuracy is critical for business utility, including long-term user retention and profitability, whereas fairness is essential for meeting legal requirements and maintaining public trust in the systems and their owners. Therefore, ML systems should be designed to enable administrators to flexibly adjust the balance between accuracy and fairness according to their operational needs.

Motivated by these demands, many researchers have developed fair ML frameworks that enable control over the fairness-accuracy trade-off through a prescribed parameter. For instance, some approaches enforce the fairness by incorporating the constraint on the permissible level of unfairness into the training objective, where the trade-off is controlled by specifying a predefined unfairness limit (Zafar et al., 2017; Celis et al., 2021). Other approaches penalize the training objective by the degree of the model unfairness, allowing the trade-off to be adjusted via a multiplicative penalty parameter (Olfat & Mintz, 2020; Bendekgey & Sudderth, 2021; Yang et al., 2023). In both cases, the resulting model is strictly tailored to the specific trade-off parameter chosen at the outset of the optimization process.

The desired trade-off between accuracy and fairness may need to be adjusted after a model has been deployed. The following example underscores the practical importance of enabling post-deployment adjustment of the accuracy–fairness trade-off.

**Example 1.1** (Regulatory Compliance in LLMs)**.** Consider a scenario involving the deployment of a Large Language Model (LLM). A company may initially set a trade-off parameter $\delta_1$ to align with existing AI guidelines. Suppose that several months after the service begins, the guidelines are updated to mandate stricter fairness standards, rendering the original setting $\delta_1$ non-compliant and necessitating an update to a new parameter $\delta_2$. However, retraining the model to accommodate this shift is often practically impossible due to the immense computational investment required. For models of this scale, retraining the entire network solely to adjust a trade-off parameter is economically and practically

---
[1]University of Tsukuba, Japan [2]RIKEN AIP, Japan. Correspondence to: Maaya Sakata <maaya@mdl.cs.tsukuba.ac.jp>, Kazuto Fukuchi <fukuchi@cs.tsukuba.ac.jp>.

*Proceedings of the 43rd International Conference on Machine Learning*, Seoul, South Korea. PMLR 306, 2026. Copyright 2026 by the author(s).

infeasible.

We refer to the ability to modify the trade-off after training as *post-hoc controllability* of the fairness-accuracy trade-off.

Existing fair machine learning approaches either lack post-hoc controllability or achieve it only at the cost of an inefficient fairness–accuracy trade-off. For example, post-processing–based fair learning algorithms achieve fairness by adjusting a trained model and therefore provide post-hoc controllability. However, they often suffer from a sub-optimal trade-off efficiency (Woodworth et al., 2017). In contrast, approaches such as in-processing methods enforce fairness during training and can achieve highly efficient trade-offs, but adapting to a new trade-off parameter typically requires costly retraining the model.

**Our contributions**  The primary goal of this paper is to develop a fair classification algorithm that enables the post-hoc controllability of the fairness-accuracy trade-off while maintaining high trade-off efficiency. Our contributions are summarized as follows:

- We theoretically analyze the accuracy improvement of post-processed classifiers when the required fairness level is relaxed and characterize the intermediate feature properties that governs trade-off efficiency. Specifically, we show that post-processed classifiers achieve more efficient trade-offs when features are more concentrated near the decision boundary of the most fair classifier. This theoretical result provides a guiding design principle for our proposed method.
- We develop a novel representation learning algorithm, *Guidance to Fairest-Boundary* (GFB), to improve the trade-off efficiency of post-processed classifiers. Motivated by our theoretical analysis, the learning objective penalizes the distance of features from the decision boundary of the most fair classifier. The resulting formulation leads to a bi-level optimization problem.
- We develop a practical gradient-based optimization procedure to solve the proposed bi-level learning problem. Specifically, we adapt the Moving-Average SOBA (MA-SOBA) (Chen et al., 2024) algorithm to our setting and address the challenge of computing gradients for the proposed learning objective.
- We conduct experiments on real-world datasets to compare our method with the existing in-processing and post-processing approaches. The results show that our method consistently outperforms the post-processing baseline by achieving more efficient fairness–accuracy trade-offs, while also attaining competitive performance relative to in-processing methods without requiring computationally expensive model retraining.[1]

---

All omitted proofs are deferred to the appendices.

## 2. Related Work

As fairness requirements or operational policies are likely to evolve continuously in practice scenarios, post-hoc controllability is essential for real-world deployment. From this perspective, post-processing methods naturally provide the post-hoc controllability, as they allow trade-off parameters to be adjusted at inference time. A representative class of approaches assigns group-specific thresholds to the prediction scores of a trained classifier to satisfy prescribed fairness criteria (Menon & Williamson, 2018; Pathiraja et al., 2023; Xian et al., 2023; Zeng et al., 2024). However, the theoretical analysis by Woodworth et al. (2017) shows that under the constraint of equalized odds, one of the fairness definition, post-processing cannot achieve optimal accuracy. This limitation is unlikely to be specific to equalized odds and may also arise under other fairness metrics.

In-processing methods can achieve efficient fairness-accuracy trade-offs by explicitly incorporating trade-off parameters into the optimization process, but they generally lack the post-hoc controllability. For example, some methods enforce fairness through optimization constraints (Zafar et al., 2017; Donini et al., 2018; Cotter et al., 2019), while others introduce penalty terms into the loss function to discourage fairness violations (Kamishima et al., 2012; Fukuchi et al., 2013; Baharlouei et al., 2024); in both of cases, the trade-off parameters are embedded in the training process. Additionally, some approaches obtain efficient trade-offs by leveraging multi-objective optimization (Liu & Vicente, 2022), where the trade-off is governed by a predefined parameter. Adapting these methods to a new trade-off parameter typically requires full retraining of the model, posing a significant barrier to flexible deployment.

Recently, Taufiq et al. (2024) applied the YOTO framework (Dosovitskiy & Djolonga, 2020) to fair machine learning, enabling efficient and post-hoc controllable trade-offs with a single model. However, YOTO-based approaches inherently require larger model capacity (Dosovitskiy & Djolonga, 2020), resulting in higher inference-time computational overhead compared to post-processing methods.

Our approach combines the strengths of both in-processing and post-processing. It learns effective feature representations such that subsequent post-processing at any target fairness level yields high trade-off efficiency. This enables efficient and flexible adjustment of the fairness-accuracy trade-off without the need for model retraining.

# 3. Preliminaries

## 3.1. Fair Classification Problem

We consider a fair binary classification problem. Let $X \in \mathcal{X}$, $A \in \{0, 1\}$, and $Y \in \{0, 1\}$ be random variables representing a feature vector, binary sensitive attribute, and binary label, respectively. A probabilistic classifier $f$ is a measurable function from $\mathcal{X} \times \{0, 1\}$ to $[0, 1]$, where the output represents the probability that the predicted label equals 1. We denote the prediction produced by $f$ by $\hat{Y}_f$, i.e., $P(\hat{Y}_f = 1 | X, A) = f(X, A)$ almost surely. Let $\mathcal{F}$ denote the set of all measurable functions $\mathcal{X} \times \mathcal{A} \rightarrow [0, 1]$. Given a set of $n$ i.i.d. observations $S = \{(x_i, a_i, y_i)\}_{i=1}^{n}$ drawn from a distribution $\mathcal{D}$, the learner's goal is to construct a family of classifiers $(f_c)_{c \in \mathcal{C}} \subseteq \mathcal{F}$ indexed by a set of trade-off parameters $\mathcal{C} \subset \mathbb{R}$ that achieve the efficient fairness-accuracy trade-offs.

Trade-off efficiency of a family of classifiers $(f_c)$ is characterized by the induced set of fairness-accuracy metrics pairs, denoted by $T((f_c)) = \{(Acc(f_c), |DDP(f_c)|\}$. Here, Acc denotes the accuracy metric, for which we adopt the standard classification accuracy; namely,

$$Acc(f_c) = P(\hat{Y}_{f_c} = Y).$$

DDP denotes the fairness metric based on demographic parity (DP) (Pedreshi et al., 2008), which deems a classifier $f \in \mathcal{F}$ fair if its predicted label is independent of the sensitive attribute. Formally, $f$ satisfies DP if

$$P(\hat{Y}_{f_c} = 1 \mid A = 1) = P(\hat{Y}_{f_c} = 1 \mid A = 0). \quad (1)$$

To quantify deviations from Equation (1), we use the difference of demographic parity (DDP) (Cho et al., 2020; Zeng et al., 2024), defined as

$$DDP(f_c) = P(\hat{Y}_{f_c} = 1 \mid A = 1) - P(\hat{Y}_{f_c} = 1 \mid A = 0).$$

A larger value of Acc (closer to 1) indicates higher accuracy, a smaller value of $|DDP(f_c)|$ (closer to 0) corresponds to greater fairness.

In this work, we aim to develop a fair learning algorithm whose resulting trade-off, $T(f_c)$, approximates the optimal trade-off $T^*(\mathcal{F})$ as closely as possible while providing post-hoc controllability. Here, post-hoc controllability refers to the ability to adapt to different trade-off parameters at inference time with a computational cost no greater than a single forward pass, without retraining the model. The optimal trade-off $T^*(\mathcal{F})$ is characterized by the Pareto front. In our context, this Pareto front consists of all achievable pairs $(Acc(f), |DDP(f)|)$ such that neither objective can be improved without degrading the other.

For notational simplicity, we henceforth write $p_a := P(A = a)$ and $\eta_a(x) := P(Y = 1 \mid X = x, A = a)$ and denote the logit of $\eta_a(x)$ by $z_a(x)$.

## 3.2. Fair Bayes-Optimal Classifier

A point on the Pareto front $T^*(\mathcal{F})$ can be obtained as the solution to the following optimization problem:

$$f_\delta^* \in \arg\max_{f \in \mathcal{F}} Acc(f) \quad \text{s.t.} \quad |DDP(f)| \leq \delta. \quad (2)$$

We refer to $f_\delta^*$ as the *fair Bayes-optimal classifier*. Under the assumption that $\eta_a(X)$ has a density on $[0, 1]$, this classifier is equivalent to

$$f_t^*(x, a) = I\left(z_a(x) > -\log \frac{p_a - (2a - 1)t}{p_a + (2a - 1)t}\right), \quad (3)$$

for an appropriately chosen $t$ (Menon & Williamson, 2018; Zeng et al., 2024), where $I(\cdot)$ denotes the indicator function. Equation (3) employs the logit representation derived from the original formulation based on the conditional class probability $\eta_a(x)$. The parameter $t$ is selected as

$$t_\delta^* = \arg\min_{t} \{|t| : |DDP(f_t^*)| \leq \delta\}. \quad (4)$$

When $t = 0$, the classifier corresponds to the unconstrained Bayes-optimal classifier. For notational convenience, we define the threshold in Equation (3) as

$$\tau_a(t) = -\log \frac{p_a - (2a - 1)t}{p_a + (2a - 1)t}. \quad (5)$$

## 3.3. FairBayes

FairBayes (Zeng et al., 2024) is a post-processing method building upon the theory of fair Bayes-optimal classifiers and uses the fairness tolerance $\delta$ described in Section 3.2 as its trade-off parameter. To realize the analytical solution $f_{t_\delta^*}^*$ in practice, FairBayes empirically estimates the unknown distributional quantities appearing in Equation (3) and substitutes them with empirical counterparts.

**Training phase: estimating the logit functions** The training phase seeks to estimate the optimal logit functions $z_a$, which are approximated using a parametric model $z_a(x; \theta_{z_a})$, where $\theta_{z_a}$ denotes the model parameters. These parameters are estimated by solving the following empirical risk minimization problem:

$$\theta_{z_a}^{ERM} = \arg\min_{\theta_{z_a}} \mathcal{L}_{pred}(\theta_{z_a}; a),$$

where, for each group $a \in \{0, 1\}$,

$$\mathcal{L}_{pred}(\theta_{z_a}; a) = \frac{1}{n_a} \sum_{i:a_i=a} \ell\big(y_i, z_a(x_i; \theta_{z_a})\big). \quad (6)$$

Here, $n_a = \sum_{i=1}^{n} I(a_i = a)$ denotes the number of samples in group $a$, and $\ell(\cdot, \cdot)$ is loss function, such as the cross-entropy loss and focal loss (Lin et al., 2020). For notational simplicity, we use $z_a^{ERM}(x)$ as shorthand for $z_a(x; \theta_{z_a}^{ERM})$.

**Algorithm 1** `PostProcess`$(z_0^{\mathrm{ERM}}, z_1^{\mathrm{ERM}}, \delta, S)$

---

**Require:** estimated logit functions $z_0^{\mathrm{ERM}}, z_1^{\mathrm{ERM}}$, trade-off parameter $\delta$
**Ensure:** estimated threshold parameter $\hat{t}_\delta$
1: Estimate group priors by $\hat{p}_a \leftarrow n_a/n$.
2: Compute $\hat{\tau}_a(t)$ and $\widehat{f}_0^{\mathrm{ERM}}$ using (7).
3: Compute $\widehat{\mathrm{DDP}}(Z(\theta_{z_a}^{\mathrm{ERM}}), 0)$ using (8).
4: **if** $\left| \widehat{\mathrm{DDP}}(Z(\theta_{z_a}^{\mathrm{ERM}}), 0) \right| \leq \delta$ **then**
5: $\quad \hat{t}_\delta(\theta_{z_a}^{\mathrm{ERM}}) \leftarrow 0.$
6: **else**
7: $\quad$ Solve Eq. (9) via binary search.
8: **end if**
9: **return** $\hat{t}_\delta(\theta_{z_a}^{\mathrm{ERM}})$

---

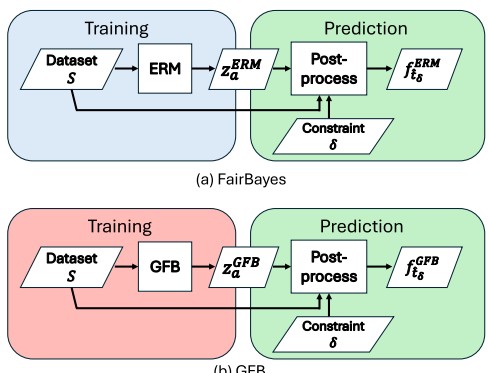

*Figure 1.* Comparison of algorithmic flows: (a) FairBayes and (b) the proposed method.

**Prediction phase: estimating the thresholds** Given a trade-off parameter $\delta$, the goal in the prediction phase is to estimate $t_\delta^*$, and consequently, the corresponding optimal classifier $f_{t_\delta^*}^*$. Algorithm 1 summarizes the steps of the entire prediction-phase procedure, which we denote by `PostProcess`$(\cdot)$.

Algorithm 1 first constructs estimates of $f_t^*$. Since estimates of the logit functions $z_a$ are already obtained in the training phase as $z_a^{\mathrm{ERM}}$, the remaining quantities to be estimated are the group-dependent thresholds $\tau_a$. Using the approximation $\hat{p}_a = n_a/n$, these thresholds are estimated (Alg. 1, lines 1–2) as

$$\hat{\tau}_0(t) = -\log \frac{\hat{p}_0 + t}{\hat{p}_0 - t}, \qquad \hat{\tau}_1(t) = -\log \frac{\hat{p}_1 - t}{\hat{p}_1 + t}. \quad (7)$$

Using these thresholds, the estimates of $f_t^*$ are then given by

$$f_t^{\mathrm{ERM}}(x, a) = I\Big( z_a(x; \theta_{z_a}^{\mathrm{ERM}}) > \hat{\tau}_a(t) \Big).$$

Next, the parameter $t_\delta^*$ is estimated through an empirical approximation of Eq. (4). Given a vector $Z \in \mathbb{R}^n$ containing the estimated logit values for all observations, DDP for classifiers induced by thresholding at $\hat{\tau}_a(t)$ is estimated as

$$\widehat{\mathrm{DDP}}(Z, t) = \frac{1}{n_1} \sum_{i:a_i=1} I\Big( Z_i > \hat{\tau}_1(t) \Big) - \frac{1}{n_0} \sum_{i:a_i=0} I\Big( Z_i > \hat{\tau}_0(t) \Big), \quad (8)$$

where $Z_i$ denotes the $i$-th component of $Z$. The estimate of $t_\delta^*$ corresponding to a given logit vector $Z$ is then obtained as

$$\hat{t}_\delta(Z) = \arg\min_t \Big\{ |t| : \big| \widehat{\mathrm{DDP}}(Z, t) \big| \leq \delta \Big\}. \quad (9)$$

Since $\widehat{\mathrm{DDP}}$ is monotone non-increasing in $t$, this optimization can be efficiently solved via binary search (Alg. 1, lines 3–9). This procedure requires at most $\mathcal{O}(\log n)$ evaluations

of $\widehat{\mathrm{DDP}}$, each of which has cost $\mathcal{O}(n)$, resulting in an overall complexity of $\mathcal{O}(n \log n)$. This is substantially more computationally efficient than retraining a model.

Once the estimate of $t_\delta^*$ via `PostProcess`$(\cdot)$ in Algorithm 1, the resulting classifier is obtained by substituting this estimate into $f_t^{\mathrm{ERM}}$. Let $Z(\theta_{z_a}^{\mathrm{ERM}}) = \big( z_{a_1}^{\mathrm{ERM}}(x_1), \ldots, z_{a_n}^{\mathrm{ERM}}(x_n) \big)^\top \in \mathbb{R}^n$ denote the logit vector for the logit functions $z_a^{\mathrm{ERM}}$. The estimate of $t_\delta^*$ is then given by $\hat{t}_\delta\big( Z(\theta_{z_a}^{\mathrm{ERM}}) \big)$, which we also denote by the shorthand $\hat{t}_\delta\big( \theta_{z_a}^{\mathrm{ERM}} \big)$. The final classifier is therefore $\widehat{f}_{\hat{t}_\delta(\theta_{z_a}^{\mathrm{ERM}})}^{\mathrm{ERM}}$.

## 4. Proposed Method

In this section, we present our novel fair classification algorithm, *Guidance to Fairest-Boundary* (GFB). Our method builds upon a theoretical analysis of the relationship between the trade-off efficiency of the post-processed classifier and the data distribution (Section 4.1). While existing results characterize the trade-off efficiency of the optimal classifier under the underlying distribution, our analysis additionally captures the trade-off efficiency achieved when the optimal classifier is applied to a different distribution.

Based on our theoretical analysis, GFB learns transformed feature representations to achieve high trade-off efficiency. Figure 1 compares the procedures of FairBayes and our proposed method. Our method maintains post-hoc controllability by adopting the same `post-process` as FairBayes during inference (right in Figure 1), while simultaneously improving the trade-off efficiency by learning appropriate latent representations during training (left bottom in Figure 1).

We also present a practical gradient-based optimization algorithm for our proposed method based on the Moving-Average SOBA (MA-SOBA) framework (Chen et al., 2024), described in Section 4.3. To adapt this framework to our

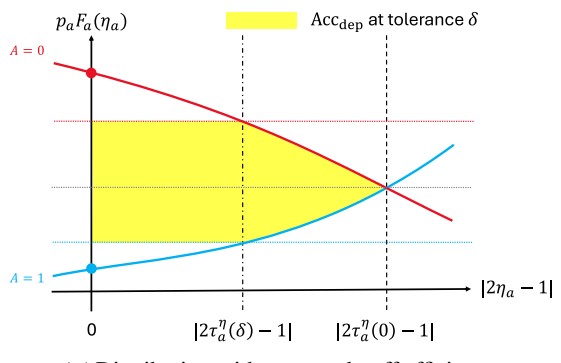

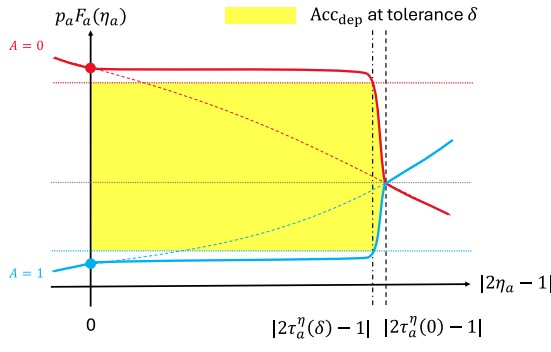

(a) Distribution with poor trade-off efficiency

(b) Distribution with good trade-off efficiency

*Figure 2.* **Effect of distributional structure on trade-off efficiency.** The horizontal and vertical axes denote $|2\eta_a - 1|$ and $p_a F_a(\eta_a)$, respectively, where $F_a(\cdot)$ denotes the cumulative distribution function conditioned by $A = a$. Yellow regions indicate the values of the $\text{Acc}_{\text{dep}}$ at tolerance $\delta$. While the figure above depicts a specific scenario where $|2\tau_0(\delta) - 1| = |2\tau_1(\delta) - 1|$, the conclusion holds true even in general cases, despite the visualization becoming more complex.

objective, we address the challenges that arise in computing gradients for our learning objective.

## 4.1. Theoretical Analysis of Trade-off Efficiency and Data Distribution

This subsection presents a theoretical result characterizing the trade-off efficiency of a post-processed classifier under changes in the data distribution. Specifically, consider a random variable corresponding a transformed feature vector $X' \in \mathcal{X}'$, and let $\tau_a^\eta(\delta) : \delta \to \mathbb{R}$ be a monotonic function of fairness tolerance $\delta$ such that the distance $|\tau_a^\eta(\delta) - 0.5|$ increases as the tolerance $\delta$ becomes more stringent. We define a post-processed classifier $f_\delta^\eta$ based on $\eta_A(X')$ by employing $\tau_a^\eta(\delta)$ as its decision threshold:

$$f_\delta^\eta(x', a) = I\big(\eta_a(x') > \tau_a^\eta(\delta)\big). \tag{10}$$

The following theorem characterizes the trade-off efficiency of the generalized classifier $f_\delta^\eta$:

**Theorem 4.1** (Accuracy dependence on the fairness tolerance $\delta$)**.** *For any classifier of the form in Equation (10), its accuracy satisfies*

$$\text{Acc}(f_\delta^\eta) = \text{Acc}(f_0^\eta) + \text{Acc}_{\text{dep}}(f_\delta^\eta),$$

*where*

$$\text{Acc}_{\text{dep}}(f_\delta^\eta) = \mathbb{E}_{X', A}\Big[I(\eta_A(X') \in \mathcal{I}_A(\delta)) \big|2\eta_A(X') - 1\big|\Big],$$

*and*

$$\mathcal{I}_a(\delta) := \Big( \min\big(\tau_a^\eta(\delta), \tau_a^\eta(0)\big), \ \max\big(\tau_a^\eta(\delta), \tau_a^\eta(0)\big)\Big].$$

Theorem 4.1 shows that the accuracy of $f_\delta^\eta$ decomposes into a $\delta$-independent term and a $\delta$-dependent term. Since the trade-off efficiency is affected solely by the $\delta$-dependent

term, $\text{Acc}_{\text{dep}}$ characterizes the trade-off efficiency. Here, $\mathcal{I}_A(\delta)$ denotes the interval such that $\eta_a(x') \in \mathcal{I}_a(\delta)$ if and only if $f_\delta^\eta(x', a) \neq f_0^\eta(x', a)$.

Importantly, this theorem applies to any classifier of the form in Equation (10). In the DDP case, it coincides with Equation (3) by setting $\tau_a^\eta(\delta) = \sigma(\tau_a(t_\delta^*))$, up to the change of input domain from $X$ to $X'$. Extensions to other fairness metrics are discussed in Appendix B.3.

Theorem 4.1 suggests that higher trade-off efficiency is achieved when the distributions of $\eta_a(X')$ are concentrated near $\tau_a^\eta(0)$, the most fair threshold, thereby increasing $\text{Acc}_{\text{dep}}$. Figure 2 illustrates the values of $\text{Acc}_{\text{dep}}$ under two representative distributions, where the areas of the yellow regions correspond to $\text{Acc}_{\text{dep}}(f_\delta^\eta)$. The red and blue lines represent $P_{X'|A=0}(\eta_0(X') \leq \tau_0^\eta(\delta))$ and $P_{X'|A=1}(\eta_1(X') \leq \tau_1^\eta(\delta))$, respectively, as functions of $\delta$ along the horizontal axis. The two distributions of $\eta_a(X')$ outside of $\mathcal{I}_a(1)$ are identical, whereas within $\mathcal{I}_a(1)$, the distribution in Figure 2b is more concentrated near $\tau_a^\eta(0)$ than that in Figure 2a. Accordingly, Figure 2 shows that the yellow area associated with Figure 2b is larger than that associated with Figure 2a. These observations suggest a key design principle for achieving superior trade-off efficiency: reshaping the distribution so that $\eta_a(X')$ concentrates near the most fair threshold $\tau_a^\eta(0)$ increases the value of $Acc_{\text{dep}}$.

## 4.2. Training Algorithm

In this subsection, we present the training algorithm for GFB. Motivated by the analyses in Section 4.1, we introduce a parametrized transformation $g_a(\cdot; \theta_{g_a}) : \mathcal{X} \to \mathcal{X}'$ from $X$ to $X'$ and optimize its parameters during training so that the resulting distribution of $\eta_a(X')$ is concentrated near $\tau_a^\eta(0)$. By integrating representation learning during training with post-processing at inference, GFB achieves superior fairness-accuracy trade-offs while remaining post-

hoc controllability.

### 4.2.1. DESIGN OF DISTRIBUTION-SHAPING LOSS $\mathcal{L}_{gen}$

We train the transformation such that the resulting logits 1) are concentrated near the most fair threshold and 2) achieve high predictive accuracy. To this end, we introduce parameters $\theta_{z_a}$ for $z_a(\cdot; \theta_{z_a})$, which estimates the logits from transformed features. Our learning objective is defined over the logits induced from parameters $\theta_{g_a}$ and $\theta_{z_a}$ and consists of two terms corresponding to these goals, defined as

$$\mathcal{L}_{gen}(\theta_{g_a}, \theta_{z_a}; a)$$
$$= (1 - \lambda)\mathcal{L}_{dist}(\theta_{g_a}, \theta_{z_a}; a) + \lambda \mathcal{L}_{pred}(\theta_{g_a}, \theta_{z_a}; a).$$

Here, $\mathcal{L}_{dist}$ and $\mathcal{L}_{pred}$ corresponds to two objectives, weighted by $\lambda$.

**Distance loss $\mathcal{L}_{dist}$**  The function $\mathcal{L}_{dist}$ encourages the logit distribution within the interval to concentrate near the most fair threshold. Let $J(\tau)$ denote the interval in the logit space corresponding to $\mathcal{I}_A(1)$ for a given threshold $\tau$, defined as $J(\tau) = \big( \min(0, \tau), \max(0, \tau) \big]$. We measure the proximity of a logit within $J(\tau)$ to the threshold $\tau$ by

$$D(z, \tau) = \begin{cases} |\tau - z|, & z \in J(\tau), \\ 0, & \text{otherwise.} \end{cases} \quad (11)$$

Based on this function, $\mathcal{L}_{dist}$ is given by

$$\mathcal{L}_{dist}(\theta_{g_a}, \theta_{z_a}; a) =$$
$$\frac{1}{n_a} \sum_{i:a_i=a} \left[ D\Big( z_a\big(g_a(x_i; \theta_{g_a}); \theta_{z_a}\big), \hat{\tau}_a\big(\hat{t}_0(\theta_{g_a}, \theta_{z_a})\big) \Big) \right],$$

where $\hat{t}_0(\theta_{g_a}, \theta_{z_a})$ is shorthand for $\hat{t}_0(Z(\theta_{g_a}, \theta_{z_a}))$, and $Z(\theta_{g_a}, \theta_{z_a}) = (\tilde{z}_1, \ldots, \tilde{z}_n)^\top \in \mathbb{R}^n$ is the induced logit vector with entries $\tilde{z}_i = z_{a_i}\big(g_{a_i}(x_i; \theta_{g_{a_i}}); \theta_{z_{a_i}}\big)$.

**Prediction loss $\mathcal{L}_{pred}$**  The function $\mathcal{L}_{pred}$ measures the prediction loss of the induced logits and is defined as

$$\mathcal{L}_{pred}(\theta_{g_a}, \theta_{z_a}; a)$$
$$= \frac{1}{n_a} \sum_{i:a_i=a} \left[ \ell\big(y_i, z_a(g_a(x_i; \theta_{g_a}); \theta_{z_a})\big) \right].$$

This term promotes high predictive accuracy of the resulting prediction model.

### 4.2.2. OVERALL FORMULATION

We now present the overall learning objective of our training algorithm. To ensure that $z_a(\cdot; \theta_{z_a})$ serves as an accurate logit predictor, we choose $\theta_{z_a}$ to minimize the predictive loss. Specifically, for given transformation parameters $\theta_{g_a}$, the selected parameter is defined as

$$\theta_{z_a}^{\text{GFB}}(\theta_{g_a}) = \arg\min_{\theta_{z_a}} \mathcal{L}_{pred}(\theta_{g_a}, \theta_{z_a}; a).$$

The transformation parameters $\theta_{g_a}$ are then learned by minimizing the function $\mathcal{L}_{gen}$ evaluated at $\theta_{z_a}^{\text{GFB}}(\theta_{g_a})$, yielding

$$\min_{\theta_{g_a}} \Phi(\theta_{g_a}) := \mathcal{L}_{gen}\Big(\theta_{g_a}, \theta_{z_a}^{\text{GFB}}(\theta_{g_a}); a\Big).$$

An equivalent bi-level optimization formulation is given by

$$\begin{aligned} \min_{\theta_{g_a}, \theta_{z_a}} & \ \mathcal{L}_{gen}(\theta_{g_a}, \theta_{z_a}; a), \\ \text{s.t. } & \theta_{z_a} = \arg\min_{\theta_{z_a}} \mathcal{L}_{pred}(\theta_{g_a}, \theta_{z_a}; a). \end{aligned} \quad (12)$$

We cannot directly apply standard gradient-based optimization algorithm, such as gradient descent, to solve Equation (12), as it is a constrained optimization problem. Nevertheless, in practice, one may wish to employ the gradient-based optimization techniques, particularly when $g_a$ and $z_a$ are modeled using deep neural networks. Accordingly, in the subsequent subsection, we present a gradient-based optimization algorithm for solving Equation (12).

### 4.3. Optimization Procedure

In this subsection, we present a gradient-based optimization algorithm for solving Equation (12), highlighting its practical applicability. We adopt MA-SOBA (Chen et al., 2024), a gradient-based bi-level optimization algorithm, to address Equation (12). However, computing the gradient of the the outer objective $\mathcal{L}_{dist}$ poses two key challenges: (i) the distance metric $D(z, \tau)$ in Equation (11) is discontinuous, and (ii) the threshold parameter $\hat{t}_0(\cdot)$ is defined as the minimizer under a discontinuous constraint of $\widehat{\text{DDP}} = 0$. To address these challenges, we introduce smooth proxy functions to approximate discontinuous components and derive the analytical gradient of $\hat{t}_0$.

**MA-SOBA**  MA-SOBA is an optimization algorithm for bi-level optimization problems that relies solely on gradients of the inner and outer objective functions, avoiding explicit Hessian inversion (Chen et al., 2024). Specifically, MA-SOBA maintains auxiliary variables that approximate the product of the inverse Hessian and the gradient. It then iteratively performs simultaneous updates of the inner, outer, and auxiliary variables using a moving-average scheme. Details, including convergence, are provided in Appendix C.

Once the proposed method is formulated as a bi-level optimization problem, we leverage the MA-SOBA framework to iteratively update the inner variable $\theta_{z_a}$, the outer variable $\theta_{g_a}$, and the auxiliary variable $w_a$. At each iteration $k$, the update directions for these variables are computed as

$$\textbf{inner: } D_{z_a}(\theta_{g_a}^k, \theta_{z_a}^k, w_a^k) = \nabla_2 \mathcal{L}_{pred}(\theta_{g_a}^k, \theta_{z_a}^k) \quad (13)$$

$$\textbf{aux: } D_{w_a}(\theta_{g_a}^k, \theta_{z_a}^k, w_a^k) \quad (14)$$
$$= \nabla_{22}^2 \mathcal{L}_{pred}(\theta_{g_a}^k, \theta_{z_a}^k) w_a^k - \nabla_2 \mathcal{L}_{gen}(\theta_{g_a}^k, \theta_{z_a}^k)$$

**outer:** $D_{g_a}(\theta_{g_a}^k, \theta_{z_a}^k, w_a^k)$ (15)
$$= \nabla_1 \mathcal{L}_{gen}(\theta_{g_a}^k, \theta_{z_a}^k) - \nabla_{12}^2 \mathcal{L}_{pred}(\theta_{g_a}^k, \theta_{z_a}^k) w_a^k.$$

Here, $\nabla_1$ and $\nabla_2$ denote the gradient operators with respect to the first and second arguments, respectively. The operators $\nabla_{12}^2$ and $\nabla_{22}^2$ correspond to differentiating with respect to the first and then second arguments, and the second and then second arguments, respectively, yielding Jacobian matrices.

Computing $\nabla_1 \mathcal{L}_{gen}$ and $\nabla_2 \mathcal{L}_{gen}$ via the chain rule requires evaluating gradients of a non-differentiable component and a component defined by the minimizer of a non-differentiable function, as discussed above. To enable gradient-based optimization, we address these gradient computation challenges.

**Smoothing the distance metric** The distance metric $D(z, \tau)$ is non-differentiable due to the discontinuity at $z = 0$. To resolve this, we approximate the indicator function $I(z \in J(\tau))$ by replacing it with a smooth function $w(z) : \mathbb{R} \to [0, 1]$, such as a sigmoid function. We then define the smooth surrogate as $\hat{D}(z, \tau) = w(z) \cdot |z - \tau|$.

**Analytical derivation of the threshold gradient** To compute the gradient of $\hat{t}_0$, we need to address both the non-continuity of $\widehat{\text{DDP}}$ and the gradient computation of a function defined as a minimizer. To handle the non-differentiability of $\widehat{\text{DDP}}$, we replace the indicator functions with a smooth, monotonically increasing function $\psi : \mathbb{R} \to \mathbb{R}$. While our focus in this work is DDP, the following derivation applies to a broad class of disparity measures. To maintain this generality, we introduce a generalized formulation that encompasses DDP. Let $\mathcal{S}_a \subseteq \{1, \ldots, n\}$ be an arbitrary subset of samples with sensitive attribute $A = a$. Further, let $\gamma_a(t)$ be a continuous and monotonic threshold function satisfying the additional technical assumptions detailed in Appendix D.2. We define the general surrogate disparity function as:

$$\phi(Z; t) = \frac{1}{|\mathcal{S}_1|} \sum_{i \in \mathcal{S}_1} \psi(Z_i - \gamma_1(t)) - \frac{1}{|\mathcal{S}_0|} \sum_{i \in \mathcal{S}_0} \psi(Z_i - \gamma_0(t)).$$
(16)

This formulation reduces to the smooth $\widehat{\text{DDP}}$ by setting $\mathcal{S}_a = \{i : a_i = a\}$ and $\gamma_a(t) = \hat{\tau}_a(t)$. Extensions to other fairness metrics are discussed in Appendix B.3.

Under this surrogate, $\hat{t}_0$ is replaced by a mapping $Z \to \arg\min_t \{|t| : \phi(Z; t) = 0\}$. The following theorem characterizes the gradient of such a function.

**Theorem 4.2** (Implicit Gradient of $t$). *Let $\phi(Z, t)$ be the surrogate disparity function defined in Equation (16). Assume that a function $\psi$ is continuously differentiable satisfying $\psi'(u) > 0$ for all $u \in \mathbb{R}$. Then, for every $Z \in \mathbb{R}^n$, there*

---

**Algorithm 2** Overall Procedure of GFB

**Require:** Dataset $S$, number of step $K$, stepsizes $\{\alpha_a^k, \beta_a^k, \gamma_a^k\}$, moving-average parameter $\rho_a^k \in (0, 1)$, trade-off parameter $\delta$.
**Ensure:** Fair classifier $f_\delta^{\text{GFB}}(x, a)$.
1: **Initialize** $\theta_{g_a}^0, \theta_{z_a}^0 / w_a^0 \leftarrow \mathbf{0}, h_a^0 \leftarrow \mathbf{0}$ for $a \in \{0, 1\}$.
2: **Training Phase**: Learning $g_a$ and $z_a$
3: **for** $k = 0, \ldots, K - 1$ **do**
4:     Sample a minibatch $B_a^k \sim S$.
5:     $v_a^k \leftarrow z_a(g_a(x; \theta_{g_a}^k); \theta_{z_a}^k)$ for $(x, y) \in B_a^k$.
6:     Compute $t^k$ from $v_0^k, v_1^k$
7:     **Outer:** $\theta_{g_a}^{k+1} \leftarrow \theta_{g_a}^k - \alpha_a^k h_a^k$. (15)
8:     **Inner:** $\theta_{z_a}^{k+1} \leftarrow \theta_{z_a}^k - \beta_a^k D_{z_a}(\theta_{g_a}^k, \theta_{z_a}^k, w_a^k)$. (13)
9:     **Aux:** $w_a^{k+1} \leftarrow w_a^k - \gamma_a^k D_{w_a}(\theta_{g_a}^k, \theta_{z_a}^k, w_a^k)$. (14)
10:     **Moving avg:** $h_a^{k+1} \leftarrow (1 - \rho_a^k) h_a^k + \rho_a^k D_{\theta_{g_a}}(\theta_{g_a}^k, \theta_{z_a}^k, w_a^k)$. (25)
11: **end for**
12: **Prediction Phase: Threshold search**
13: $z_a^{\text{GFB}} \leftarrow z_a(g_a(\cdot; \theta_{g_a}^K); \theta_{z_a}^K)$
14: $\hat{t}_\delta \leftarrow \texttt{PostProcess}(z_0^{\text{GFB}}, z_1^{\text{GFB}}, \delta, S)$
15: Compute $\hat{\tau}_a(\hat{t}_\delta)$.
16: **return** $f_\delta^{\text{GFB}}(x, a) = I(z_a^{\text{GFB}}(x_a) > \hat{\tau}_a(\hat{t}_\delta))$.

---

*exists a unique parameter $t(Z)$ satisfying*

$$\phi(Z, t(Z)) = 0.$$

*Moreover, $t(Z)$ is differentiable with respect to $Z$, and its gradient is given by*

$$\nabla_1 t(Z) = -\frac{\nabla_1 \phi(Z, t(Z))}{\nabla_2 \phi(Z, t(Z))}.$$

By replacing non-smooth components with smooth surrogates and leveraging gradient computation in Theorem 4.2, we can solve Equation (12) using MA-SOBA. Algorithm 2 summarizes the complete pipeline of the proposed approach, spanning both the training and inference phases.

## 5. Experiments

To demonstrate the effectiveness of our method, we conduct experiments on several real-world datasets and compare it with existing in-processing and post-processing methods.

### 5.1. Experimental Setup

**Comparison methods** In our experiments, we compare the proposed method with several competitive baselines: EPO (Mahapatra & Rajan, 2020) (multi-objective optimization), FairBiNN (Yazdani-Jahromi et al., 2024) (bi-level optimization), YOTO (Taufiq et al., 2024) (in-processing), and FairBayes (Zeng et al., 2024) (post-processing). EPO

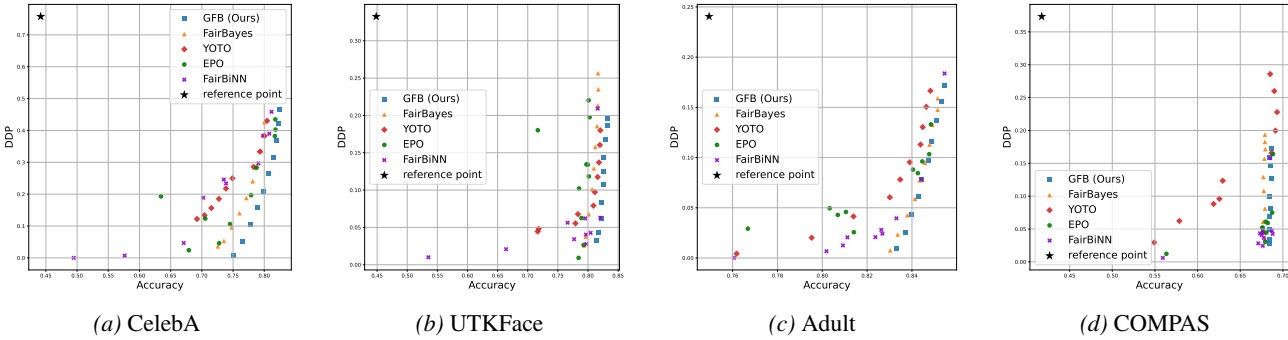

*Figure 3.* Fairness-accuracy trade-off curves for each dataset. The horizontal and vertical axes represent accuracy and DDP.

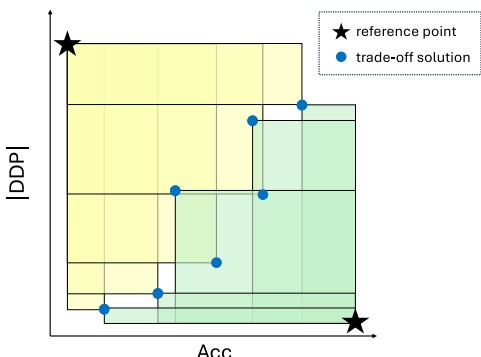

*Figure 4.* Illustration of the standard HV and inverted HV. The yellow region represents the standard HV, which evaluates favorable trade-off coverage, where larger values are better. The green region represents the inverted HV, which evaluates the extent of poorly performing trade-off solutions, where smaller values are better.

and FairBiNN lacks the post-hoc controllability, whereas both YOTO and FairBayes provide the post-hoc controllability.

**Datasets**   Experiments were conducted on four datasets: two image datasets, CelebA (Liu et al., 2015) and UTK-Face (Zhang et al., 2017), and two tabular datasets, Adult (Kohavi, 1996) and COMPAS (Angwin et al., 2016). The target labels and sensitive attributes for each dataset are summarized in Table 3 of Section E.2.

**Evaluation metrics**   We evaluate classification accuracy (Acc) and the absolute demographic parity difference (|DDP|) as the accuracy and fairness metrics, respectively. The reported values are averages over 5 runs. To quantify the fairness-accuracy trade-off, we use two hypervolume-based metrics: standard HV and inverted HV, as illustrated in Figure 4. Both metrics measure an area relative to a reference point, but place the reference point in opposite directions. The standard HV uses a reference point at the worst corner, corresponding to low Acc and high |DDP|.

The inverted HV uses a reference point at the best corner, corresponding to high Acc and low |DDP|. Details of the computation are provided in Section E.3.

Larger standard HV indicates better coverage of favorable trade-off solutions, while smaller inverted HV indicates fewer poorly performing solutions. A method with high standard HV but also high inverted HV cannot be considered efficient over the entire trade-off curve, since it produces poor solutions for some trade-off parameters despite obtaining favorable ones for others. Together, larger standard HV and smaller inverted HV indicate more efficient and stable trade-off behavior. Although both metrics are sensitive to the choice of reference point, the ranking of methods remains largely unchanged under different settings; see Section E.5.3 for details.

### 5.2. Results

Figures 3a–3d show the fairness-accuracy trade-off of each method. The horizontal axis represents Acc, with higher values toward the right, while the vertical axis represents DDP, with greater fairness toward the bottom. Thus, a trade-off curve closer to the bottom-right region indicates a more efficient fairness-accuracy trade-off. Each point corresponds to a specific trade-off parameter. For YOTO, FairBayes, and GFB, a single model is trained once on the training dataset and then evaluated on the testing dataset under 10 different trade-off parameters in a post-hoc manner. Since EPO and FairBiNN do not provide post-hoc controllability, each reported point for these methods corresponds to a separately trained model. Table 1 and Table 2 summarize the mean HV values and inverted HV values computed from these trade-offs, respectively. Additional statistics are provided in Appendix E.5.2.

First, we compare our method with post-hoc controllable methods, FairBayes and YOTO. As shown in Figure 3, except for the comparison with FairBayes on Adult, GFB is located below or to the right of those of FairBayes and YOTO across all datasets. This indicates that GFB gener-

*Table 1.* Comparison of Hypervolume. Values are reported as mean $\pm$ standard deviation.

|  | EPO | FairBiNN | YOTO | FairBayes | GFB (Ours) |
|---|---|---|---|---|---|
| CelebA | $0.7733 \pm 0.060$ | $0.6919 \pm 0.045$ | $0.6068 \pm 0.146$ | $0.7257 \pm 0.026$ | $\mathbf{0.8198 \pm 0.005}$ |
| UTKFace | $0.8455 \pm 0.024$ | $\mathbf{0.8876 \pm 0.024}$ | $0.7481 \pm 0.078$ | $0.8202 \pm 0.020$ | $0.8581 \pm 0.025$ |
| Adult | $0.7237 \pm 0.072$ | $0.7553 \pm 0.055$ | $0.6492 \pm 0.047$ | $0.7850 \pm 0.041$ | $\mathbf{0.8007 \pm 0.038}$ |
| COMPAS | $0.8191 \pm 0.060$ | $\mathbf{0.8409 \pm 0.074}$ | $0.6442 \pm 0.133$ | $0.7598 \pm 0.084$ | $0.8164 \pm 0.038$ |

*Table 2.* Comparison of inverted Hypervolume. Values are reported as mean $\pm$ standard deviation. Lower values are better.

|  | EPO | FairBiNN | YOTO | FairBayes | GFB (Ours) |
|---|---|---|---|---|---|
| CelebA | $0.4563 \pm 0.180$ | $0.3829 \pm 0.057$ | $0.3223 \pm 0.228$ | $0.2622 \pm 0.017$ | $\mathbf{0.2268 \pm 0.003}$ |
| UTKFace | $0.4443 \pm 0.383$ | $0.2579 \pm 0.060$ | $0.1558 \pm 0.058$ | $0.1556 \pm 0.016$ | $\mathbf{0.1010 \pm 0.019}$ |
| Adult | $0.4801 \pm 0.054$ | $0.3507 \pm 0.035$ | $0.4314 \pm 0.076$ | $0.2529 \pm 0.027$ | $\mathbf{0.2514 \pm 0.028}$ |
| COMPAS | $0.2207 \pm 0.036$ | $0.2155 \pm 0.048$ | $0.3661 \pm 0.109$ | $0.1724 \pm 0.032$ | $\mathbf{0.1398 \pm 0.030}$ |

ally achieves lower DDP at comparable accuracy, or higher accuracy at comparable DDP, resulting in a more efficient fairness-accuracy trade-off. Consistent with this observation, our method attains higher HV and lower inverted HV than both methods across all datasets, indicating that GFB achieves more efficient trade-offs while also avoiding poorly performing trade-off points. To achieve this improvement, GFB employs a bi-level training procedure, which slightly increases the training cost compared with FairBayes and YOTO. We report the wall-clock training time and the cost of post-hoc trade-off adjustment in Appendix E.5.1.

Next, we compare our method with the in-processing method EPO. As shown in Table 1, GFB achieves higher HV than EPO on CelebA, UTKFace, and Adult, whereas EPO obtains higher HV on COMPAS. This shows that GFB outperforms EPO in terms of HV on most datasets, while EPO has an advantage on COMPAS. However, as shown in Figure 3, the trade-off curve of GFB lies to the right of, or close to, that of EPO across all datasets including COMPAS, indicating that GFB achieves comparable or higher accuracy at similar DDP levels. Furthermore, GFB achieves lower inverted HV than EPO across all datasets, suggesting that GFB produces fewer poorly performing trade-off points. Overall, GFB is competitive with EPO in trade-off performance while additionally providing post-hoc controllability that EPO does not offer.

Next, we compare our method with another in-processing method, FairBiNN. On CelebA and Adult, the trade-off curves of GFB are located to the right of, or close to, those of FairBiNN, and GFB achieves higher HV on both datasets. These results suggest that GFB attains a more efficient fairness-accuracy trade-off than FairBiNN on CelebA and Adult. On UTKFace and COMPAS, FairBiNN achieves higher HV than GFB, indicating its advantage under the HV metric. However, the curve-level comparison shows a more nuanced picture. Although the trade-off curves of FairBiNN extends further downward, GFB is often located to the right of, or close to, FairBiNN within the vertical range where both methods have solutions; this tendency is particularly visible on UTKFace. Moreover, GFB achieves lower inverted HV than FairBiNN across all datasets, indicating that FairBiNN can obtain highly favorable trade-off points that increase HV while producing poorly performing points for some trade-off parameters. In contrast, GFB avoids such poor points and maintains a more stable trade-off curve overall.

## 6. Conclusion

In this paper, we propose a novel fair classification method that provides the post-hoc controllability while achieving high trade-off efficiency. Our algorithm combines representation learning during training with post-processing at inference. Experiments on real-world datasets demonstrated the effectiveness of our approach.

We also note several limitations and future directions. First, although the framework supports several fairness metrics, it does not cover all notions; extending it to equalized odds remains an important direction. Second, our current framework is restricted to binary sensitive attributes and binary labels. Extending it to multiclass classification and intersectional groups defined by multiple sensitive attributes is an important future direction. However, such extensions require substantial technical development, as fair Bayes-optimal classifiers in these settings no longer admit simple thresholding structures (Xian et al., 2023). Addressing these extensions would further broaden the applicability of our framework.

## Acknowledgements

This work was partly supported by JSPS KAKENHI Grant Numbers JP26K02874 and JP23H00483.

## Impact Statement

This paper presents work whose goal is to advance the field of machine learning. There are many potential societal consequences of our work, none of which we feel must be specifically highlighted here.

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

# A. Extended Related Work

To achieve fairness in machine learning systems, numerous methods have been proposed that intervene at different stages of the learning pipeline to mitigate or eliminate bias. These approaches are commonly categorized into three groups.

Pre-processing methods aim to improve the fairness of predictions by adjusting the training data. They are motivated by the understanding that a primary source of model unfairness is the inherent bias present in the training data. Therefore, models trained on debiased data are expected to yield fair predictions. For example, (Kamiran & Calders, 2012) proposed a method that removes bias by modifying the labels in a subset of training data. However, in such approaches that directly modify the data itself, increasing fairness causes a discrepancy from the original one and a substantial loss of accuracy. For this reason, such methods struggle to achieve the efficient trade-off. Moreover, when the desired user's preferred balance between fairness and accuracy changes, the data must be reprocessed and the model retrained, these methods do not allow post-hoc control of the trade-off.

In-processing methods achieve fairness by directly incorporating fairness criteria into the model optimization process. Such approaches include methods that explicitly introduce fairness constraints into the optimization problem (Zafar et al., 2017; Donini et al., 2018; Cotter et al., 2019), those that add penalty terms for fairness violations to the loss function (Kamishima et al., 2012; Fukuchi et al., 2013; Baharlouei et al., 2024). Additionally, multi-objective optimization techniques that jointly optimize accuracy and fairness (Liu & Vicente, 2022) or bi-level optimization (Yazdani-Jahromi et al., 2024). These methods have the advantage of achieving an efficient trade-off, as each trade-off parameter such as reference vectors or weights can be individually optimized. However, they require the model to be retrained whenever the preference changes. Furthermore, the inclusion of constraints in the loss function often complicates the optimization and increases the computational cost. As a result, in-processing methods also lack post-hoc controllability.

Post-processing methods improve predictive fairness by applying post-hoc modification to the outputs of a trained model. A representative approach involves adjusting predictions by setting group-specific thresholds for the scores produced by a trained classifier to satisfy given fairness requirements (Menon & Williamson, 2018; Pathiraja et al., 2023; Xian et al., 2023; Zeng et al., 2024). Because these methods decouple model training from trade-off parameters, they can efficiently adapt to changes in trade-off parameter without retraining the model, which is a key advantage. However, Woodworth et al. (2017) showed that under the constraint of equalized odds, one of the fairness definition, post-processing cannot achieve optimal accuracy. This limitation is unlikely to be specific to equalized odds and may also arise under other fairness metrics. Similar post-processing ideas are extended to regression problems. For demographic parity constrained regression, Chzhen & Schreuder (2022) fully characterized the fairness–risk trade-off in a minimax sense. Building on this line of work, (Fukuchi, 2025) further showed that, under demographic parity, the entire family of minimax-optimal predictors can be generated via post-processing from a single meta-optimal predictor. In contrast, for classification problems, a complete characterization of the demographic parity trade-off remains open, with existing results limited to special settings (Zhao & Gordon, 2022).

Recently, Taufiq et al. (2024) applied the YOTO framework (Dosovitskiy & Djolonga, 2020) to fair machine learning, enabling efficient and post-hoc controllable trade-offs with a single model. By explicitly conditioning the model on the trade-off parameter, YOTO allows a single model to approximate the entire Pareto frontier without retraining separate models for different trade-off setting. However, this flexibility comes at the cost of increased model complexity. In particular, YOTO-based approaches inherently require substantially larger model capacity to realize their full potential. Empirical results reported in Dosovitskiy & Djolonga (2020) show that, in order for a single YOTO model to match the performance of models trained separately for individual trade-off parameters, the network typically needs to be approximately twice as wide as those fixed-weight models. As a consequence, despite avoiding repeated training, YOTO incurs higher inference-time computational cost compared to post-processing methods.

# B. Extended Details for Section 3

## B.1. Pareto Front

The set $T^*$ is characterized by the concept of the Pareto front. In our context, the Pareto front consists of all achievable pairs $(\mathrm{Acc}(f), |\mathrm{Dis}(f)|)$ that are not dominated by any other solution. Here, $\mathrm{Dis}(f)$ denotes a disparity measure associated with the corresponding fairness metric, where values closer to zero indicate higher fairness.

We say that a solution $f'$ is dominated by another solution $f$, denoted by $f' \preceq f$, if

$$\mathrm{Acc}(f) > \mathrm{Acc}(f') \quad \text{and} \quad |\mathrm{Dis}(f)| \leq |\mathrm{Dis}(f')| \qquad \text{or} \qquad \mathrm{Acc}(f) \geq \mathrm{Acc}(f') \quad \text{and} \quad |\mathrm{Dis}(f)| < |\mathrm{Dis}(f')|.$$

Following (Xu & Strohmer, 2023), the Pareto front is defined as

$$T^*(\mathcal{F}) = \left\{ \left( \mathrm{Acc}(f), |\mathrm{Dis}(f)| \right) \mid f \in \mathcal{F}, \; \nexists f' \in \mathcal{F} \text{ such that } f \preceq f' \right\},$$

where $\mathcal{F}$ denotes the set of all measurable functions.

## B.2. Fairness Metrics

In the main body, we mainly focus on Demographic Parity (DP). However, several other fairness metrics are also commonly used. In this subsection, we introduce the fairness metrics, in addition to DP, that are covered by our theoretical analysis.

**Equal Opportunity (EOp) (Hardt et al., 2016)** EOp requires a classifier $f_c$ to achieve the same true positive rate across sensitive groups:

$$P(\hat{Y}_{f_c} = 1 | Y = 1, A = 1) = P(\hat{Y}_{f_c} = 1 | Y = 1, A = 0). \tag{17}$$

To quantify deviation from Equation (17), the difference of equal opportunity (DEOp) is commonly used and is defined as

$$\mathrm{DEOp}(f_c) = P(\hat{Y}_{f_c} = 1 | Y = 1, A = 1) - P(\hat{Y}_{f_c} = 1 | Y = 1, A = 0).$$

A smaller value of $|\mathrm{DEOp}(f_c)|$, closer to 0, indicates greater fairness.

**Predictive Equality (PE) (Corbett-Davies et al., 2017)** PE requires a classifier $f_c$ to achieve the same false positive rate across sensitive groups:

$$P(\hat{Y}_{f_c} = 1 | Y = 0, A = 1) = P(\hat{Y}_{f_c} = 1 | Y = 0, A = 0). \tag{18}$$

To quantify deviation from Equation (18), the difference of predictive equality (DPE) is commonly used and is defined as

$$\mathrm{DPE}(f_c) = P(\hat{Y}_{f_c} = 1 | Y = 0, A = 1) - P(\hat{Y}_{f_c} = 1 | Y = 0, A = 0).$$

A smaller value of $|\mathrm{DPE}(f_c)|$, closer to 0, indicates greater fairness.

## B.3. Fair Bayes-Optimal Classifier

As discussed in Section 3.2, the classifier in Eq. (2) is originally defined using the conditional class probability $\eta_a(x) = P(Y = 1 \mid X = x, A = a)$. Under the assumption that $\eta_a(X)$ has a density on $[0, 1]$, fair Bayes-optimal classifiers for the fairness metrics considered in this work can be written as group-dependent threshold rules (Menon & Williamson, 2018; Zeng et al., 2024):

$$f_t^{\mathrm{Dis}}(x, a) = I \left( \eta_a(x) > \tau_a^{\mathrm{Dis}}(t) \right), \tag{19}$$

where $\mathrm{Dis} \in \{\mathrm{DP}, \mathrm{EOp}, \mathrm{PE}\}$ denotes the fairness metric, and $\tau_a^{\mathrm{Dis}}(t)$ is a group-dependent threshold controlled by the scalar parameter $t$. When $t = 0$, the DP classifier reduces to the unconstrained Bayes-optimal rule with decision threshold $1/2$, which achieves the highest possible accuracy.

In our framework, we reformulate this classifier in terms of logits $z_a(x) = \log(\eta_a(x)/(1 - \eta_a(x)))$. Equivalently, Equation (19) can be written as

$$f_t^{\mathrm{Dis}}(x, a) = I \left( z_a(x) > \gamma_a^{\mathrm{Dis}}(t) \right), \quad \gamma_a^{\mathrm{Dis}}(t) := \log \frac{\tau_a^{\mathrm{Dis}}(t)}{1 - \tau_a^{\mathrm{Dis}}(t)}.$$

Zeng et al. (2024) show that fair Bayes-optimal classifiers for other fairness metrics, such as EOp and PE, also admit analogous threshold-based forms. The corresponding thresholds are summarized below.

For DP, the threshold is

$$\tau_a^{\mathrm{DP}}(t) = \frac{1}{2} + \frac{(2a - 1)t}{2p_a}, \quad p_a = P(A = a). \tag{20}$$

For Equal Opportunity (EOp), the threshold is

$$\tau_a^{\mathrm{EOp}}(t) = \frac{p_{a,1}}{2p_{a,1} - (2a - 1)t}, \quad p_{a,1} = P(A = a, Y = 1). \tag{21}$$

Therefore, the corresponding logit threshold is

$$\gamma_a^{\text{EOP}}(t) = -\log\left(1 - \frac{(2a-1)t}{p_{a,1}}\right). \tag{22}$$

For Predictive Equality (PE), the threshold is

$$\tau_a^{\text{PE}}(t) = \frac{p_{a,0} + (2a-1)t}{2p_{a,0} + (2a-1)t}, \quad p_{a,0} = P(A = a, Y = 0). \tag{23}$$

Therefore, the corresponding logit threshold is

$$\gamma_a^{\text{PE}}(t) = -\log\left(\frac{p_{a,0}}{p_{a,0} + (2a-1)t}\right). \tag{24}$$

For each fairness metric, the parameter $t$ is chosen so that the corresponding disparity constraint is satisfied. Specifically, for a given tolerance $\delta$, $t$ is selected to achieve $\arg\min_t\{|t| : |\text{Dis}(f_t^{\text{Dis}})| \le \delta\}$.

## C. Moving-Average SOBA (MA-SOBA)

### C.1. Comprehensive Framework

MA-SOBA (Chen et al., 2024) is a fully single-loop algorithm for solving stochastic bilevel optimization problems that builds on the Stochastic Bilevel Algorithm (SOBA) (Dagréou et al., 2022). Their goal is to solve the following optimization problem, such that $g$ and $f$ represent the lower-level and upper-level functions, respectively

$$\min_{x\in\mathcal{X}} \Phi(x) := f(x, y^*(x)) \quad \text{s.t.} \quad y^*(x) = \arg\min_y g(x, y).$$

We first review the SOBA algorithm and its limitations, and then describe how MA-SOBA addresses these challenges.

SOBA introduce an auxiliary variable to approximate the product of the Hessian and a gradient, and simultaneously update the inner $y$, outer $x$, and auxiliary variables $z$ using SGD steps. Let $x^k$, $y^k$, and $z^k$ denote the values of the respective variables at iteration $k$. Their update rules are given as follows:

$$
\begin{aligned}
y^{k+1} &= y^k - \beta_k \nabla_2 g(x^k, y^k) \\
z^{k+1} &= z^k - \gamma_k \Big\{ \nabla_{22}^2 g(x^k, y^*(x^k)) z^k - \nabla_2 f(x^k, y^*(x^k)) \Big\} \\
&\approx z^k - \gamma_k \Big\{ \nabla_{22}^2 g(x^k, y^k) z^k - \nabla_2 f(x^k, y^k) \Big\} \\
x^{k+1} &= x^k - \alpha_k \Big\{ \nabla_1 f(x^k, y^*(x^k)) - \nabla_{12}^2 g(x^k, y^*(x^k)) z^*(x^k) \Big\} = x^k - \alpha_k \nabla\Phi(x^k) \\
&\approx x^k - \alpha_k \Big\{ \nabla_1 f(x^k, y^k) - \nabla_{12}^2 g(x^k, y^k) z^k \Big\}
\end{aligned}
$$

Here, $\alpha_k$, $\beta_k$, and $\gamma_k$ denote the step sizes. Since the inner variable $y^k$ is updated using only a single SGD step at each iteration, it generally does not coincide with the exact solution $y^*(x^k)$. As a consequence, the stochastic gradient used in the update of the auxiliary variable $z$ is biased. Similarly, the hypergradient $\nabla\Phi(x^k)$ is also subject to bias.

To mitigate the bias in hypergradient estimation, MA-SOBA incorporates a moving-average mechanism into the update rules. Specifically, MA-SOBA introduces a sequence of variables $\{h^k\}$ that aggregates past biased stochastic hypergradients. The update is given by

$$h^{k+1} = (1 - \theta_k)h^k + \theta_k \Big\{ \nabla_1 f(x^k, y^k) - \nabla_{12}^2 g(x^k, y^k) z^k \Big\}. \tag{25}$$

Here, $\theta_k$ denotes the weight parameter of the moving average. MA-SOBA then replaces the update rule for $x$ with one that uses the averaged hypergradient: $x^{k+1} = x^k - \alpha_k h^k$, thereby mitigating the bias induced by inexact inner updates.

## C.2. Convergence Analysis

We provide a brief discussion on the convergence of the MA-SOBA optimizer used in our framework. Following the theoretical analysis by Chen et al. (2024), MA-SOBA is guaranteed to converge under the following standard assumptions:

1. First-order Lipschitz continuity of the outer objective and second-order Lipschitz continuity of the inner objective.
2. Strong convexity of the inner objective.
3. Boundedness of the gradient at the optimal solution of the inner problem.

Assumptions 1 and 3 can be satisfied by choosing a doubly differentiable $\psi$ with bounded first and second derivatives. In our experiments, we use the sigmoid function as $\psi$, which satisfies these conditions. Assumption 2 requires the ERM objective to be strongly convex with respect to the model parameters, which is generally not satisfied in practice. However, we note that local strong convexity around local minima may suffice for convergence, as gradient-based optimization typically remains within such local region. The local strong convexity can be further encouraged in practice, e.g., by incorporating weight decay.

# D. Proofs

## D.1. Proof of Theorem 4.1

**Theorem D.1** (Accuracy dependence on the fairness tolerance $\delta$). *The accuracy of classifiers of the form in Eq. (10) satisfies*

$$\mathrm{Acc}(f_\delta^\eta) = \mathrm{Acc}(f_0^\eta) + \mathrm{Acc}_{\mathrm{dep}}(f_\delta^\eta),$$

*where*

$$\mathrm{Acc}_{\mathrm{dep}}(f_\delta^\eta) = \mathbb{E}_{X',A}\Big[I(\eta_A(X') \in \mathcal{I}_A(\delta))\, \big|2\eta_A(X') - 1\big|\Big],$$

*and*

$$\mathcal{I}_a(\delta) := \Big( \min\big(\tau_a^\eta(\delta),\, \tau_a^\eta(0)\big),\ \max\big(\tau_a^\eta(\delta),\, \tau_a^\eta(0)\big)\Big].$$

*Proof.* $\mathcal{I}_A(\delta)$ denotes the interval such that $\eta_a(x') \in \mathcal{I}_a(\delta)$ if and only if $f_\delta^\eta(x', a) \neq f_0^\eta(x', a)$. Specifically, When $\delta$ is sufficiently large, $\tau_a^\eta = 0.5$ holds, and thus $\mathcal{I}_A(1) = \big(\min(0.5, \tau_a^\eta(0)),\ \max(0.5, \tau_a^\eta(0))\big]$.

For a fixed $(X', A)$, we consider the probability that the prediction of the Bayes-optimal classifier, $\hat{Y}_{f_1^\eta}$, coincides with the true label $Y$: $P\left(\hat{Y}_{f_1^\eta} = Y \mid X', A\right)$. If $\eta_A(X') > 0.5$, the classifier predicts $\hat{Y}_{f_1^\eta} = 1$, and this probability equals $P(Y = 1 \mid X', A) = \eta_A(X')$. If $\eta_A(X') \leq 0.5$, the classifier predicts $\hat{Y}_{f_1^\eta} = 0$, and the probability equals $\mathbb{P}(Y = 0 \mid X', A) = 1 - \eta_A(X')$. Hence, the probability that the prediction matches the true label is given by

$$P\left(\hat{Y}_{f_1^\eta} = Y \mid X', A\right) = \max(\eta_A(X'),\, 1 - \eta_A(X')) = \frac{1 + |2\eta_A(X') - 1|}{2}.$$

Now, consider the classifier $f_\delta^\eta$ with a shifted threshold $\eta_A(X') > \tau_a^\eta(\delta)$. The conditional probability $P\left(\hat{Y}_{f_\delta^\eta} = Y \mid X', A\right)$ depends on whether the prediction of $f_\delta^\eta$ coincides with that of Bayes-optimal classifier $f_1^\eta$. If $\hat{Y}_{f_\delta^\eta} = \hat{Y}_{f_1^\eta}$, the conditional probability is $\max(\eta_A(X'), 1 - \eta_A(X'))$. On the other hand, if $\hat{Y}_{f_\delta^\eta} \neq \hat{Y}_{f_0^\eta}$, it becomes

$$P\left(\hat{Y}_{f_\delta^\eta} = Y \mid X', A\right) = \min(\eta_A(X'), 1 - \eta_A(X')) = \frac{1 - |2\eta_A(X') - 1|}{2}.$$

Therefore, the accuracy of most fair classifier $f_0^\eta$ can be written as

$$\mathrm{Acc}(f_0^\eta) = \mathbb{E}_{X',A}\left[\frac{1 + |2\eta_A(X') - 1|}{2} I(\eta_A(X') \notin \mathcal{I}_A(1))\right] + \mathbb{E}_{X',A}\left[\frac{1 - |2\eta_A(X') - 1|}{2} I(\eta_A(X') \in \mathcal{I}_A(1))\right].$$

For a given tolerance $\delta$, the predicted label $\hat{Y}_{f_\delta^\eta}$ agree with $\hat{Y}_{f_1^\eta}$ on the interval $\mathcal{I}_A(\delta)$. Accordingly, the accuracy of $f_\delta^\eta$ satisfies

$$\mathrm{Acc}(f_\delta^\eta) = \mathrm{Acc}(f_0^\eta) - \mathbb{E}_{X',A}\left[\frac{1 - |2\eta_A(X') - 1|}{2} \cdot I(\eta_A(X') \in \mathcal{I}_A(\delta))\right]$$

$$+ \mathbb{E}_{X',A}\left[\frac{1 + |2\eta_A(X') - 1|}{2} \cdot I(\eta_A(X') \in \mathcal{I}_A(\delta))\right]$$
$$= \mathrm{Acc}(f_0^\eta) + \mathbb{E}_{X',A}\left[|2\eta_A(X') - 1| \cdot I(\eta_A(X') \in \mathcal{I}_A(\delta))\right]$$
$$= \mathrm{Acc}(f_0^\eta) + \mathrm{Acc}_{\mathrm{dep}}(f_\delta^\eta).$$

$\square$

**Corollary D.2** (Applicability to Standard Fairness Metrics). *The threshold functions $\tau_a^\eta(t)$ corresponding to DP, EOp, and PE, given in Equations (20), (21) and (23), satisfy the conditions required in Theorem 4.1. Specifically, for each fairness metric, the threshold $\tau_a^\eta(\delta)$ varies monotonically with the fairness tolerance $\delta$, and the distance $|\tau_a^\eta(\delta) - 0.5|$ increases as the fairness constraint becomes more stringent, i.e., as $\delta$ decreases. Moreover, the corresponding fair Bayes-optimal classifier can be written in the form of Equation (10). Consequently, Theorem 4.1 holds for DP, EOp, and PE.*

*Proof.* The classifiers for DP, EOp, and PE all admit the threshold form in Equation (10), with thresholds given in Equations (20), (21) and (23). For each metric, differentiating the corresponding threshold functions shows that $\tau_1^\eta$ and $\tau_0^\eta$ vary monotonically in opposite directions with respect to the scalar threshold parameter. Moreover, at the unconstrained point, the thresholds reduce to $0.5$, and moving toward a stricter fairness constraint shifts the thresholds away from $0.5$. Hence, the conditions required in Theorem 4.1 are satisfied, and the theorem applies to DP, EOp, and PE. $\square$

### D.2. Proof of Theorem 4.2 (Implicit Gradient of $t$)

Before stating our main result regarding the existence and uniqueness of the optimal parameter $t$, we formalize the necessary assumptions on the mapping function $\psi$ and the threshold functions $\tau_a(t)$.

**Assumption D.3** (Strict Monotonicity of $\psi$). The mapping function $\psi$ is strictly increasing such that $\psi'(u) > 0$ for any $u \in \mathbb{R}$.

Let $\mathcal{T} \subset \mathbb{R}$ be the maximal open interval consisting of all parameters $t$ for which the threshold functions $\gamma_a(t)$ are strictly well-defined and yield finite real values for both $a \in \{0, 1\}$. Formally, we define this feasible domain as:

$$\mathcal{T} = \{t \in \mathbb{R} \mid \gamma_0(t) \in \mathbb{R} \text{ and } \gamma_1(t) \in \mathbb{R}\} = (p^-, p^+),$$

where the boundaries $p^-$ and $p^+$ naturally arise from the domain restrictions inherent to the definitions of $\gamma_a(t)$. Over this interval $\mathcal{T}$, we assume the following properties for the threshold functions:

**Assumption D.4** (Continuity). $\gamma_0(t)$ and $\gamma_1(t)$ are continuous on $\mathcal{T}$.

**Assumption D.5** (Monotonicity). $\gamma_1(t)$ is monotonically increasing and $\gamma_0(t)$ is monotonically decreasing with respect to $t$.

**Assumption D.6** (Sufficient Separation). There exist parameters within the open interval $\mathcal{T}$ that completely separate the shifted scores $Z_i - \gamma_a(t)$ between the two groups. Specifically, there exist $t_1, t_2 \in \mathcal{T}$ such that for all $i$ with $a_i = 1$ and $j$ with $a_j = 0$, the following inequalities hold:

$$Z_i - \gamma_1(t_1) < Z_j - \gamma_0(t_1) \quad \text{and} \quad Z_i - \gamma_1(t_2) > Z_j - \gamma_0(t_2).$$

With these assumptions in place, we guarantee the existence and uniqueness of the threshold shifting parameter.

**Lemma D.7** (Existence and uniqueness of $t$). *Let $\phi$ be the surrogate fairness metric defined in Equation (16). Suppose Assumptions D.3, D.4, D.5, and D.6 hold. Then, for any score vector $Z \in \mathbb{R}^n$, there exists a unique parameter $t^* \in \mathcal{T}$ that satisfies $\phi(Z, t^*) = 0$.*

*Proof.*

- **Uniqueness.** Fix any $Z \in \mathbb{R}^n$. Since $\gamma_1(t)$ is strictly increasing and $\gamma_0(t)$ is strictly decreasing on $\mathcal{T}$, and $\psi'(u) > 0$ for all $u \in \mathbb{R}$, $\psi(Z_i - \gamma_1(t))$ is strictly decreasing in $t$ and $\psi(Z_i - \gamma_0(t))$ is strictly increasing in $t$. Therefore, $\phi(Z, t)$ is strictly decreasing in $t$, and hence the equation $\phi(Z, t) = 0$ has at most one solution.
- **Existence.** Fix any $Z \in \mathbb{R}^n$. We evaluate the limits of $\phi(Z, t)$ at the boundaries of $\mathcal{T}$.

(i) We first show that $\phi(Z, t) < 0$ for some $t \in \mathcal{T}$. Under the above assumptions, there exists a parameter $t_1 \in \mathcal{T}$ that satisfies $Z_i - \gamma_1(t_1) < Z_j - \gamma_0(t_1)$ for all $i \in \mathcal{S}_1$ and for all $j \in \mathcal{S}_0$. Since $\psi$ is a strictly monotonically increasing function, this order is strictly preserved: $\psi(Z_i - \gamma_1(t_1)) < \psi(Z_j - \gamma_0(t_1))$. Taking the average over each group, we have

$$\frac{1}{|\mathcal{S}_1|} \sum_{i \in \mathcal{S}_1} \psi(Z_i - \gamma_1(t_1)) < \frac{1}{|\mathcal{S}_0|} \sum_{j \in \mathcal{S}_0} \psi(Z_j - \gamma_0(t_1)).$$

Consequently, at this $t = t_1$, we obtain

$$\phi(Z, t_1) = \frac{1}{|\mathcal{S}_1|} \sum_{i \in \mathcal{S}_1} \psi(Z_i - \gamma_1(t_1)) - \frac{1}{|\mathcal{S}_0|} \sum_{j \in \mathcal{S}_0} \psi(Z_j - \gamma_0(t_1)) < 0.$$

This confirms that there exists some $t \in \mathcal{T}$ such that $\phi(Z, t) < 0$.
(ii) We then show that $\phi(Z, t) > 0$ for some $t \in \mathcal{T}$. Under the above assumptions, there exists a parameter $t_2 \in \mathcal{T}$ that satisfies $Z_i - \gamma_1(t_2) > Z_j - \gamma_0(t_2)$ for all $i \in \mathcal{S}_1$ and for all $j \in \mathcal{S}_0$. Since $\psi$ is a strictly monotonically increasing function, this order is strictly preserved: $\psi(Z_i - \gamma_1(t_2)) > \psi(Z_j - \gamma_0(t_2))$. Taking the average over each group, we have

$$\frac{1}{|\mathcal{S}_1|} \sum_{i \in \mathcal{S}_1} \psi(Z_i - \gamma_1(t_2)) > \frac{1}{|\mathcal{S}_0|} \sum_{j \in \mathcal{S}_0} \psi(Z_j - \gamma_0(t_2)).$$

Consequently, at this $t = t_2$, we obtain

$$\phi(Z, t_2) = \frac{1}{|\mathcal{S}_1|} \sum_{i \in \mathcal{S}_1} \psi(Z_i - \gamma_1(t_2)) - \frac{1}{|\mathcal{S}_0|} \sum_{j \in \mathcal{S}_0} \psi(Z_j - \gamma_0(t_2)) > 0.$$

This confirms that there exists some $t \in \mathcal{T}$ such that $\phi(Z, t) < 0$.
Since $\phi(Z, t)$ is continuous on $\mathcal{T}$ and there exist $t_1, t_2 \in \mathcal{T}$ such that $\phi(Z, t_1) > 0$ and $\phi(Z, t_2) < 0$), the Intermediate Value Theorem guarantees the existence of at least one $t \in \mathcal{T}$ such that $\phi(Z, t) = 0$.

$\square$

**Proposition D.8** (Derivation of the local gradient $\nabla_1 t(Z)$). *Suppose Assumptions D.3, D.4, D.5, and D.6 hold. The function $\phi : \mathbb{R}^n \times \mathbb{R} \to \mathbb{R}$ is continuously differentiable. For any $Z_0 \in \mathbb{R}^n$, let $t_0 = t(Z_0)$ be the unique solution guaranteed by Lemma D.7, so that $\phi(Z_0, t_0) = 0$. Since $\psi'(u) > 0$, we have $\nabla_2 \phi(Z_0, t_0) \neq 0$. Therefore, by the Implicit Function Theorem, there exist a neighborhood $U \subset \mathbb{R}^n$ of $Z_0$ and a neighborhood $V \subset \mathbb{R}$ of $t_0$, and a unique continuously differentiable local function $\tilde{t} : U \to V$ such that*

$$\phi(Z, \tilde{t}(Z)) = 0 \qquad \forall Z \in U.$$

*Moreover, its gradient is given by*

$$\nabla_1 \tilde{t}(Z) = -\frac{\nabla_1 \phi(Z, \tilde{t}(Z))}{\nabla_2 \phi(Z, \tilde{t}(Z))}. \tag{26}$$

*Proof.* By assumption, the function $\phi$ is continuously differentiable. For any $Z_0 \in \mathbb{R}^n$, let $t_0 = t(Z_0)$ be the unique solution such that $\phi(Z_0, t_0) = 0$, whose existence and uniqueness are guaranteed by Lemma D.7. Moreover, since $\psi'(u) > 0$, we have $\nabla_2 \phi(Z_0, t_0) \neq 0$. Therefore, all the conditions of the Implicit Function Theorem are satisfied. As a result, there exists a neighborhood $U$ of $Z_0$ and a unique continuously differentiable local function $\tilde{t} : U \to \mathbb{R}$ such that $\phi(Z, \tilde{t}(Z)) = 0$ for all $Z \in U$, and its gradient is given by Equation (26). $\square$

Having established the general theoretical guarantees, we now demonstrate that the threshold functions derived from common fairness metrics, specifically Demographic Parity (DP), Equal Opportunity (EOp), and Predictive Equality (PE), naturally satisfy Assumptions D.4, D.5, and D.6.

**Corollary D.9** (Applicability to Standard Fairness Metrics). *The threshold functions $\gamma_a(t)$ corresponding to DP, EOp, and PE satisfy Assumptions D.4, D.5, and D.6. Consequently, by Lemma D.7, for $Z \in \mathbb{R}^n$, there exists a unique optimal parameter $t^* \in \mathcal{T}$ that strictly satisfies the fairness constraint $\phi(Z, t^*) = 0$ for each of these metrics.*

*Proof.* We prove this corollary by verifying that the threshold functions for each fairness metric satisfy the required assumptions.

**Demographic Parity.** For DP, the logit threshold functions $\gamma_a^{\mathrm{DP}}(t)$ are given in Equation (5). The feasible domain is $\mathcal{T} = (-p, p)$, where $p = \min(p_0, p_1)$. This domain ensures that the arguments of the logarithms are strictly positive.

- **Continuity (Assumption D.4).** For any $t \in \mathcal{T}$, the log arguments in $\gamma_a^{\mathrm{DP}}(t)$ are positive. Hence, $\gamma_a^{\mathrm{DP}}(t)$ is continuous on $\mathcal{T}$ as a composition of continuous functions.
- **Monotonicity (Assumption D.5).** The derivatives of the threshold functions with respect to $t$ are

$$\nabla_1 \gamma_0^{\mathrm{DP}}(t) = -\frac{2p_0}{(p_0 - t)(p_0 + t)} < 0, \quad \nabla_1 \gamma_1^{\mathrm{DP}}(t) = \frac{2p_1}{(p_1 - t)(p_1 + t)} > 0.$$

Since $t^2 < p^2 \leq p_a^2$ for all $t \in \mathcal{T}$, the denominators are positive. Therefore, $\gamma_1^{\mathrm{DP}}(t)$ is strictly increasing and $\gamma_0^{\mathrm{DP}}(t)$ is strictly decreasing, satisfying Assumption D.5.
- **Sufficient Separation (Assumption D.6).** Assume without loss of generality that $p = p_1 \leq p_0$.
(i) As $t \to p$ from the left, $\gamma_1^{\mathrm{DP}}(t) \to \infty$. Thus, $Z_i - \gamma_1^{\mathrm{DP}}(t) \to -\infty$ for all $i \in \mathcal{S}_1$. On the other hand, the group-0 terms do not diverge to $-\infty$; in particular, $\gamma_0^{\mathrm{DP}}(t)$ remains finite if $p_1 < p_0$, and diverges to $-\infty$ if $p_1 = p_0$. In either case, there exists $t_1 \in \mathcal{T}$ such that

$$Z_i - \gamma_1^{\mathrm{DP}}(t_1) < Z_j - \gamma_0^{\mathrm{DP}}(t_1) \quad \text{for all } i \in \mathcal{S}_1, \ j \in \mathcal{S}_0.$$

(ii) As $t \to -p$ from the right, $\gamma_1^{\mathrm{DP}}(t) \to -\infty$. Hence, $Z_i - \gamma_1^{\mathrm{DP}}(t) \to \infty$ for all $i \in \mathcal{S}_1$. Meanwhile, the group-0 terms do not diverge to $+\infty$; if $p_1 < p_0$, $\gamma_0^{\mathrm{DP}}(t)$ remains finite, and if $p_1 = p_0$, $\gamma_0^{\mathrm{DP}}(t) \to \infty$. Therefore, there exists $t_2 \in \mathcal{T}$ such that

$$Z_i - \gamma_1^{\mathrm{DP}}(t_2) > Z_j - \gamma_0^{\mathrm{DP}}(t_2) \quad \text{for all } i \in \mathcal{S}_1, \ j \in \mathcal{S}_0.$$

Thus, Assumption D.6 holds for DP.

**Equal Opportunity.** For EOp, the logit threshold functions $\gamma_a^{\mathrm{EOp}}(t)$ are given in Equation (22). The feasible domain is $\mathcal{T} = (-p_{0,1}, p_{1,1})$, which guarantees that the logarithm arguments are positive.

- **Continuity (Assumption D.4).** Throughout $\mathcal{T}$, the log arguments in $\gamma_a^{\mathrm{EOp}}(t)$ are strictly positive. Therefore, $\gamma_a^{\mathrm{EOp}}(t)$ is continuous on $\mathcal{T}$.
- **Monotonicity (Assumption D.5).** Differentiating the two threshold functions gives

$$\nabla_1 \gamma_0^{\mathrm{EOp}}(t) = -\frac{1}{p_{0,1} + t} < 0, \quad \nabla_1 \gamma_1^{\mathrm{EOp}}(t) = \frac{1}{p_{1,1} - t} > 0.$$

Since $-p_{0,1} < t < p_{1,1}$, both denominators are positive. Hence, $\gamma_1^{\mathrm{EOp}}(t)$ is strictly increasing and $\gamma_0^{\mathrm{EOp}}(t)$ is strictly decreasing, as required.
- **Sufficient Separation (Assumption D.6).** (i) As $t \to p_{1,1}$ from the left, $\gamma_1^{\mathrm{EOp}}(t) \to \infty$, while $\gamma_0^{\mathrm{EOp}}(t)$ converges to a finite value. Consequently, $Z_i - \gamma_1^{\mathrm{EOp}}(t) \to -\infty$ for all $i \in \mathcal{S}_1$, whereas $Z_j - \gamma_0^{\mathrm{EOp}}(t)$ remains bounded for all $j \in \mathcal{S}_0$. Thus, there exists $t_1 \in \mathcal{T}$ such that

$$Z_i - \gamma_1^{\mathrm{EOp}}(t_1) < Z_j - \gamma_0^{\mathrm{EOp}}(t_1) \quad \text{for all } i \in \mathcal{S}_1, \ j \in \mathcal{S}_0.$$

(ii) As $t \to -p_{0,1}$ from the right, $\gamma_0^{\mathrm{EOp}}(t) \to \infty$, while $\gamma_1^{\mathrm{EOp}}(t)$ remains finite. Therefore, $Z_j - \gamma_0^{\mathrm{EOp}}(t) \to -\infty$ for all $j \in \mathcal{S}_0$, and there exists $t_2 \in \mathcal{T}$ such that

$$Z_i - \gamma_1^{\mathrm{EOp}}(t_2) > Z_j - \gamma_0^{\mathrm{EOp}}(t_2) \quad \text{for all } i \in \mathcal{S}_1, \ j \in \mathcal{S}_0.$$

Thus, Assumption D.6 holds for EOp.

**Predictive Equality.** For PE, the logit threshold functions $\gamma_a^{\mathrm{PE}}(t)$ are given in Equation (24). The feasible domain is $\mathcal{T} = (-p_{1,0}, p_{0,0})$, which ensures that the logarithm arguments are strictly positive.

- **Continuity (Assumption D.4).** For every $t \in \mathcal{T}$, the denominator $p_{a,0} + (2a-1)t$ is positive. Hence, $\gamma_a^{\mathrm{PE}}(t)$ is continuous on $\mathcal{T}$.

- **Monotonicity (Assumption D.5).** The derivatives are

$$\nabla_1 \gamma_0^{\text{PE}}(t) = -\frac{1}{p_{0,0} - t} < 0, \quad \nabla_1 \gamma_1^{\text{PE}}(t) = \frac{1}{p_{1,0} + t} > 0.$$

Since $-p_{1,0} < t < p_{0,0}$, both denominators are positive. Therefore, $\gamma_1^{\text{PE}}(t)$ is strictly increasing and $\gamma_0^{\text{PE}}(t)$ is strictly decreasing.

- **Sufficient Separation (Assumption D.6).** (i) As $t \to p_{0,0}$ from the left, $\gamma_0^{\text{PE}}(t) \to -\infty$, while $\gamma_1^{\text{PE}}(t)$ remains finite. Thus, $Z_j - \gamma_0^{\text{PE}}(t) \to \infty$ for all $j \in \mathcal{S}_0$. Hence, there exists $t_1 \in \mathcal{T}$ such that

$$Z_i - \gamma_1^{\text{PE}}(t_1) < Z_j - \gamma_0^{\text{PE}}(t_1) \quad \text{for all } i \in \mathcal{S}_1, \; j \in \mathcal{S}_0.$$

(ii) As $t \to -p_{1,0}$ from the right, $\gamma_1^{\text{PE}}(t) \to -\infty$, while $\gamma_0^{\text{PE}}(t)$ remains finite. Consequently, $Z_i - \gamma_1^{\text{PE}}(t) \to \infty$ for all $i \in \mathcal{S}_1$. Therefore, there exists $t_2 \in \mathcal{T}$ such that

$$Z_i - \gamma_1^{\text{PE}}(t_2) > Z_j - \gamma_0^{\text{PE}}(t_2) \quad \text{for all } i \in \mathcal{S}_1, \; j \in \mathcal{S}_0.$$

Thus, Assumption D.6 holds for PE.

Since all three fairness metrics satisfy the continuity, monotonicity, and sufficient separation assumptions, the proof is complete. $\qquad\square$

## E. Experiments

### E.1. Comparison methods

**FairBayes** FairBayes (Zeng et al., 2024) is a post-processing method derived from the theory of fair Bayes-optimal classifiers. It controls the balance between fairness and accuracy without requiring retraining by identifying the decision boundaries corresponding to specific trade-off parameters $\delta$.

To realize the fair bayes-optimal classifier $f_{t_\delta^*}^*$ in practice, FairBayes empirically estimates the unknown distributional quantities in Equation (3).

**Training phase**: FairBayes learns group-dependent logit functions $z_a^{\text{ERM}}$ by empirical risk minimization with cross-entropy loss, and the learned logits are fixed thereafter.

**Prediction phase**: Group-dependent thresholds $\tau_a(t_\delta)$ are applied to the logits, where $t_\delta$ is chosen to satisfy $|\text{DDP}| \le \delta$. Using empirical estimates of group priors $\hat{p}_a = n_a/n$, the thresholds are given by Equation (7). The parameter $\hat{t}_\delta$ is obtained by solving Equation (9).

**EPO (Exact Pareto Optimal search).** EPO (Mahapatra & Rajan, 2020) is a multi-objective optimization method that controls the descent direction across multiple objectives in order to obtain a specific solution on the Pareto front corresponding to a given trade-off parameter.

Since EPO is not originally proposed in the context of the trade-off between fairness and accuracy, we adapt it to this setting by defining task-specific objective functions. Specifically, we introduce two objective functions: $\mathcal{L}_{acc}$ for prediction accuracy and $\mathcal{L}_{ddp}$ for fairness. $\mathcal{L}_{acc}$ follows the same definition as in Eq. (6), while $\mathcal{L}_{ddp}$ is defined as a smooth approximation of DDP. Concretely, the indicator function in DDP is approximated by a scaled sigmoid function $\sigma(\cdot)$, and we define

$$\mathcal{L}_{ddp}(f) = \left| \frac{1}{n_1} \sum_{i;a=1} \sigma(k\, f(x_i, 1)) - \frac{1}{n_0} \sum_{i;a=0} \sigma(k\, f(x_i, 0)) \right|.$$

The trade-off between these objectives is controlled by a reference vector $v$. Let $v_{acc}$ and $v_{ddp}$ denote the components of the reference vector for the two objectives. EPO performs optimization by updating the model in a search direction that satisfies the weighted condition $v_{acc}\mathcal{L}_{acc} = v_{fair}\mathcal{L}_{ddp}$ and minimizes the weighted loss

$$\mathcal{L}_w(f) = v_{acc}(f)\, \mathcal{L}_{acc}(f) + v_{ddp}(f)\, \mathcal{L}_{ddp}(f).$$

*Table 3.* Summary of datasets, target labels, and sensitive features used in our experiments.

|  | CelebA | UTKFace | Adult | COMPAS |
|---|---|---|---|---|
| Target label ($Y$) | Attractive | Age $\geq 30$ | Income $\geq 50K$ | 2-year recidivism |
| Sencitive Attrbute ($A$) | Gender | Gender | Gender | Race (Caucasian and others) |

**FairBiNN (Fair Bilevel Neural Network)** FairBiNN (Yazdani-Jahromi et al., 2024) is a bilevel optimization framework for obtaining solutions on the fairness-accuracy trade-off corresponding to a given trade-off parameter.

FairBiNN treats the accuracy loss $\mathcal{L}_{acc}$ as the upper-level objective and the fairness loss $\mathcal{L}_{ddp}$ as the lower-level objective. We define $\mathcal{L}_{acc}$ and $\mathcal{L}_{ddp}$ in the same way as in the EPO formulation. The optimization problem is formulated as

$$\min_{\theta_{acc}} \mathcal{L}_{acc}\big(f(\theta_{acc}, \theta_{ddp}^*)\big) \quad \text{s.t.} \quad \theta_{ddp}^* \in \arg\min_{\theta_{ddp}} \mathcal{L}_{ddp}\big(f(\theta_{acc}, \theta_{ddp})\big).$$

Here, $\theta_{\text{acc}}$ and $\theta_{\text{ddp}}$ denote the parameters of the accuracy and fairness layers, respectively. FairBiNN partitions a single neural network into accuracy and fairness layers, which are optimized separately according to their respective objectives. During training, $\theta_{\text{acc}}$ and $\theta_{\text{ddp}}$ are updated alternately at each mini-batch.

The trade-off between fairness and accuracy is controlled by a scaling parameter $\eta \in \mathbb{R}$ that scales the fairness loss during the fairness update. Different trade-off solutions are obtained by varying $\eta$, where larger values place greater emphasis on fairness. Consequently, FairBiNN requires training a separate model for each trade-off solution.

**YOTO (You Only Train Once).** YOTO (Taufiq et al., 2024) is an in-processing method applied the original You Only Train Once framework (Dosovitskiy & Djolonga, 2020) to fair machine learning, aiming for the entire Pareto front with a single neural network. Following the terminology in (Taufiq et al., 2024), we refer to their adaptation as YOTO throughout this paper.

YOTO incorporates the trade-off parameter $\lambda$ directly as an input to the model via Feature-wise Linear Modulation (FiLM) (Perez et al., 2018). In this architecture, a FiLM layer applies an affine transformation to a given intermediate feature vector $h$:

$$\text{FiLM}(h \mid \lambda) = \gamma(\lambda) \odot h + \beta(\lambda),$$

where the scale $\gamma(\lambda)$ and shift $\beta(\lambda)$ are generated by a hypernetwork conditioned on $\lambda$. This mechanism allows the model to dynamically adapt its behavior to any given value of $\lambda$.

In our experiments, following YOTO framework, we optimizes a $\lambda$-conditioned objective function:

$$\mathcal{L}_{yoto}(f, \lambda) = \mathcal{L}_{acc}\big(f(\cdot; \lambda)\big) + \lambda \mathcal{L}_{ddp}\big(f(\cdot; \lambda)\big).$$

where $f(\cdot; \lambda)$ denotes a classifier conditioned on $\lambda$. The accuracy loss $\mathcal{L}_{\text{acc}}$ and the fairness loss $\mathcal{L}_{\text{ddp}}$ are defined in the same way as in the EPO formulation. To ensure the model learns to represent classifiers across the entire trade-off range, the parameter $\lambda$ is sampled at each iteration from a log-uniform distribution over $[10^{-6}, 10]$.

### E.2. Datasets

We summarize the target labels and sensitive attributes for each dataset in Table 3.

**CelbeA dataset** The CelebA dataset (Liu et al., 2015) consists of 202,599 face images annotated with 40 binary facial attributes. In our experiments, we use "Attractive" as the prediction target. The sensitive attribute is gender.

**UTKFace dataset** The UTKFace dataset (Zhang et al., 2017) contains over 20,000 face images labeled with age, gender, and race. We define the prediction task as determining whether an individual's age is 30 or above. The sensitive attribute is gender.

**Adult dataset** The Adult dataset (Kohavi, 1996) is a tabular dataset comprising demographic and occupational attributes such as age, profession, and education level. The prediction task is to determine whether an individual's annual income exceeds $50,000$. The sensitive attribute considered in this work is gender.

**COMPAS dataset** The COMPAS dataset (Angwin et al., 2016) is a tabular dataset used for recidivism risk prediction in the criminal justice system. It contains features such as age, race, and prior criminal history. The prediction task is to determine whether a defendant will reoffend within two years. The sensitive attribute is race, which we binarize into Caucasian and non-Caucasian groups.

### E.3. Metric

The fairness-accuracy trade-off efficiency is quantified using the hypervolume (HV) (Zitzler & Thiele, 1998) over Acc and |DDP|. In this setting, the objective space is two-dimensional. Before defining the standard HV and the inverted HV used in our experiments, we first introduce a general hypervolume function. For an arbitrary trade-off set $T$ and reference point $r$, the hypervolume function is defined as the area of the union of rectangles spanned by $r$ and each solution in $T$ (Badar et al., 2024):

$$HV(T, r) = \Lambda_2 \left( \bigcup_{i=1}^{|T|} [r, \ q(f_i)] \right),$$

where $q(f_i) = (\text{Acc}(f_i), |\text{DDP}(f_i)|)$ denotes the $i$-th solution in trade-off set $T$ and $|T|$ denotes the number of solutions in $T$. Here, $[r, q(f_i)]$ is the axis-aligned rectangle defined by $r$ and $q(f_i)$, and $\bigcup_{i=1}^{|T|} [r, q(f_i)]$ denotes the union of these rectangles. The operator $\Lambda_2(\cdot)$ denotes the two-dimensional Lebesgue measure, which corresponds to the area of the region.

In our experiments, we set the reference point following the procedure described in (Jovanovic et al., 2022), which builds on (Ishibuchi et al., 2018). We first define the reference point for standard HV, denoted by $r_{\text{stn}} = (r_{\text{acc}}, r_{\text{ddp}})$. Let $T_{j,s}$ denote the non-dominated solutions of the trade-off set obtained by algorithm $j$ with seed $s$. The objective values of all solutions in $\bigcup_{j,s} T_{j,s}$ are normalized to the range $[0, 1]$ based on the extreme solutions observed across all algorithms and seeds. The reference point for the standard HV is then defined as

$$r_{\text{acc}} = -\frac{1}{N-1}, \qquad r_{\text{ddp}} = 1 + \frac{1}{N-1}. \tag{27}$$

Here, $N = \max_{j,s} |T_{j,s}|$ denotes the maximum number of non-dominated solutions among all compared algorithms and seeds. Thus, the reference point is obtained by shifting one unit of size $1/(N-1)$ beyond the nadir point of the normalized objective space. The standard HV of a trade-off set $T$ is then defined as $HV(T, r_{\text{stn}})$.

Next, we define the reference point for inverted HV, denoted by $r_{\text{inv}} = (r'_{\text{acc}}, r'_{\text{ddp}})$. Let $\bar{T}_{j,s}$ denote the set of non-Pareto-front, i.e., dominated, solutions obtained by algorithm $j$ with seed $s$. We normalize these solutions using the same normalization procedure as above. Let $N' = \max_{j,s} |\bar{T}_{j,s}|$ denote the maximum number of dominated solutions among all compared algorithms and seeds. The reference point for the inverted HV is then defined as

$$r'_{\text{acc}} = 1 + \frac{1}{N'-1}, \qquad r'_{\text{ddp}} = -\frac{1}{N'-1}.$$

Accordingly, the reference point is obtained by shifting one unit of size $1/(N'-1)$ beyond the ideal corner of the normalized objective space, i.e., high accuracy and low DDP. The inverted HV of a trade-off set $T$ is then defined as $HV(T, r_{\text{inv}})$.

In the main experiments, each trade-off curve is evaluated using 10 trade-off parameters; therefore, both $N-1$ and $N'-1$ are at most 9. For the sensitivity analysis, we vary $N-1$ and $N'-1$ over 30 logarithmically spaced values from 2 to 80 and examine how the rankings of the methods change in Section E.5.3.

### E.4. Implementation Details

#### E.4.1. LOSS FUNCTION

For all methods, we adopt the Focal Loss (Lin et al., 2020) as the loss function $\ell$ to address the inherent class imbalance in datasets. Originally proposed for dense object detection to reduce the dominance of easy-to-classify background examples, the Focal Loss is defined as

$$\text{FL}(p_t) = -(1 - p_t)^\gamma \log(p_t),$$

where $p_t$ is the model's estimated probability for the ground-truth label, defined as

$$p_t = \begin{cases} p, & \text{if } y = 1, \\ 1 - p, & \text{otherwise,} \end{cases}$$

with $p$ being the model's estimated probability for the class with label $y = 1$. By introducing the focusing parameter $\gamma$, the loss effectively down-weights the contribution from easy samples and directs the optimization toward hard, underrepresented examples. Note that $\gamma = 0$ reduces to standard cross-entropy.

### E.4.2. TRADE-OFF PARAMETER SELECTION

For a fair comparison, we evaluate the trade-offs obtained under ten different trade-off parameters for each method.

For the proposed method and FairBayes, the trade-off is controlled by the fairness tolerance $\delta$, and the corresponding thresholds are determined using a holdout set. Specifically, we first compute the DDP at the Bayes-optimal threshold 0, denoted by $\delta_{\max}$, and then uniformly sample ten fairness tolerances $\delta$ from the interval $[0, \delta_{\max}]$.

For EPO, trade-off parameters are the reference vectors. Following (Mahapatra & Rajan, 2020), we use ten reference vectors that are evenly spaced in angle between $(v_{\mathrm{acc}}, v_{\mathrm{ddp}}) = (1, 0)$ and $(0, 1)$.

For YOTO, trade-off parameters are the weighting parameter $\lambda$. We first compute the DDP obtained at $\lambda = 0$ and denote it by $\delta_{\max}$. We then uniformly sample ten target fairness tolerances $\delta$ from the interval $[0, \delta_{\max}]$. For each target $\delta$, we estimate the corresponding $\lambda_\delta$ via a binary search on a logarithmic scale over the range $\lambda \in [10^{-6}, 10]$, such that the resulting classifier attains a DDP value close to the target tolerance $\delta$. Different trade-off points are obtained by performing inference with the classifier conditioned on $\lambda_\delta$.

### E.4.3. NETWORK ARCHITECTURE

For image datasets, we use a ResNet-18 backbone pre-trained on ImageNet, followed by a two-layer multilayer perceptron (MLP). The MLP classifier consists of two fully connected layers with ReLU activation, mapping the ResNet feature representation to a scalar logit. In the proposed framework, the ResNet-18 backbone is treated as the distribution transformation $g$, while the MLP serves as the classification head $f$. Conversely, baseline methods treat the entire network as a single classifier. For YOTO, we replace the standard MLP with a FiLM-conditioned MLP to enable trade-off conditioning. Following (Taufiq et al., 2024), the hypernetwork used to generate the FiLM parameters $\gamma(\lambda)$ and $\beta(\lambda)$ is implemented as a four-layer MLP. For FairBiNN, we adopt the same overall network architecture and apply the layer-partitioning strategy of Yazdani-Jahromi et al. (2024). Specifically, the early and intermediate blocks of the ResNet-18 backbone are treated as the first accuracy component, the final residual block group of ResNet-18 is treated as the fairness component, and the subsequent MLP classifier is treated as the second accuracy component.

For tabular datasets, we utilize TARTE (Kim et al., 2025), a Transformer-based foundation model, as a fixed feature extractor. Following the architecture in (Kim et al., 2025), we feed the readout token's representation into an MLP classifier. We use a 7-layer MLP for Adult and a 5-layer MLP for COMPAS. In our method, the final two layers of the MLP act as the classification head $f$, with the preceding layers serving as the transformation module $g$. For baseline comparisons, the entire MLP is treated as $f$, and for YOTO, it is replaced with a FiLM-conditioned architecture. The hypernetwork is implemented as a two-layer MLP, consistent with the configuration in (Taufiq et al., 2024). For FairBiNN, we use the same TARTE-based architecture and partition the trainable MLP layers following (Yazdani-Jahromi et al., 2024). Specifically, we split the trainable MLP after TARTE into three parts: early accuracy layers, intermediate fairness layers, and a final accuracy head. The fairness layers correspond to the later transformation layers placed before the final accuracy head, while the remaining trainable layers are treated as accuracy layers.

### E.4.4. HYPERPARAMETER TUNING

For the proposed method and FairBayes, we select the model that achieves the largest HV on the validation set. To compute HV, we first evaluate the DDP at the Bayes-optimal threshold 0 and denote it as $\delta_{\max}$. We then construct a trade-off curve by uniformly sampling 50 fairness tolerances $\delta \in [0, \delta_{\max}]$. To ensure a consistent comparison across epochs, the reference point for the validation HV is fixed at the worst-case point $(r_{\mathrm{acc}}, r_{\mathrm{ddp}}) = (0, 1)$. For EPO, we select the model that minimizes the weighted loss on the validation set. For YOTO, we select the model that minimizes the validation loss. A single model is conditioned on a weighting parameter $\lambda$, which does not directly correspond to the fairness tolerance $\delta$. While it is in principle possible to evaluate validation HV by fixing a set of $\lambda$ values, this would require performing inference for each $\lambda$ over the entire validation set at every epoch, resulting in a substantially higher computational cost.

*Table 4.* Wall-clock time (seconds) on the **CelebA** dataset for all methods (averaged over 5 seeds). * per parameter for EPO/FairBiNN/YOTO, 10 parameters for FairBayes/GFB.

|  | EPO | FairBiNN | YOTO | FairBayes | GFB (Ours) |
|---|---|---|---|---|---|
| Train | $4.58 \times 10^3$ | $2.24 \times 10^3$ | $1.77 \times 10^3$ | $1.78 \times 10^3$ | $2.98 \times 10^3$ |
| Adaptation | same as above | same as above | 0 | $5.10 \times 10^{-2}$ | $5.16 \times 10^{-2}$ |
| Inference* | $5.72 \times 10^0$ | $5.88 \times 10^0$ | $5.03 \times 10^0$ | $5.52 \times 10^0$ | $5.55 \times 10^0$ |

*Table 5.* Wall-clock time (seconds) on the **UTKFace** dataset for all methods (averaged over 5 seeds). * per parameter for EPO/FairBiNN/YOTO, 10 parameters for FairBayes/GFB.

|  | EPO | FairBiNN | YOTO | FairBayes | GFB (Ours) |
|---|---|---|---|---|---|
| Train | $1.49 \times 10^3$ | $7.67 \times 10^2$ | $5.75 \times 10^2$ | $5.65 \times 10^2$ | $9.61 \times 10^2$ |
| Adaptation | same as above | same as above | 0 | $9.66 \times 10^{-3}$ | $9.38 \times 10^{-3}$ |
| Inference* | $1.52 \times 10^0$ | $1.61 \times 10^0$ | $1.56 \times 10^0$ | $1.43 \times 10^0$ | $1.66 \times 10^0$ |

### E.4.5. COMPUTING ENVIRONMENT

The core software stack consisted of Python 3.12.3 and PyTorch 2.6.0a0. For GPU acceleration, we utilized CUDA 12.8 with NVIDIA driver version 535.183.01. All computations and model training were performed on an NVIDIA A100-SXM4-40GB GPU.

## E.5. Additional Experimental Results

### E.5.1. TRAINING COST

We evaluated computational time on all datasets, as shown in Table 4 to 7. In these Tables, "Train" reports the training time for 100 epochs, excluding hyperparameter tuning, "Adaptation" denotes the additional computation time required when changing the trade-off parameter (threshold estimation for FairBayes/GFB, retraining for EPO), and "Inference" denotes the time required to produce predictions on the test set. For FairBayes and GFB, the training time includes one forward pass on a holdout set used for threshold estimation to compute logits.

The training time of GFB is approximately 1.2-1.7 times longer than that of FairBayes. However, we believe that this moderate increase in training cost does not undermine the overall benefits of our method. In particular, GFB consistently outperforms FairBayes and YOTO across all datasets with smaller standard deviations in most cases. This suggests that GFB provides a more stable trade-off performance. Moreover, after training, GFB retains the lightweight post-hoc adjustment of FairBayes and achieves shorter inference time than YOTO. These results suggest that, despite the additional training cost, GFB remains practical for settings where the desired fairness–accuracy trade-off may change after deployment.

Compared with in-processing methods, GFB consistently requires less training time than EPO. On the other hand, GFB takes approximately 1.04–1.33 times longer to train than FairBiNN. However, unlike EPO and FairBiNN, which require retraining when the trade-off parameter changes, GFB only needs lightweight threshold adaptation after training. As shown in Tables 4 to 7, this adaptation is substantially faster than retraining and enables efficient post-hoc control of the fairness-accuracy trade-off.

Considering the consistent improvements over FairBayes and YOTO discussed above, together with the practical benefit of post-hoc controllability, we regard the additional training cost of GFB as acceptable.

*Table 6.* Wall-clock time (seconds) on the **Adult** dataset for all methods (averaged over 5 seeds). * per parameter for EPO/FairBiNN/YOTO, 10 parameters for FairBayes/GFB.

|  | EPO | FairBiNN | YOTO | FairBayes | GFB (Ours) |
|---|---|---|---|---|---|
| Train | $1.15 \times 10^2$ | $1.02 \times 10^2$ | $7.89 \times 10^1$ | $7.51 \times 10^1$ | $1.07 \times 10^2$ |
| Adaptation | same as above | same as above | 0 | $1.61 \times 10^{-2}$ | $1.60 \times 10^{-2}$ |
| Inference* | $4.82 \times 10^{-1}$ | $6.36 \times 10^{-1}$ | $5.19 \times 10^{-1}$ | $5.06 \times 10^{-1}$ | $5.11 \times 10^{-1}$ |

*Table 7.* Wall-clock time (seconds) on the **COMPAS** dataset for all methods (averaged over 5 seeds). * per parameter for EPO/FairBiNN/YOTO, 10 parameters for FairBayes/GFB.

|  | EPO | FairBiNN | YOTO | FairBayes | GFB (Ours) |
|---|---|---|---|---|---|
| Train | $6.71 \times 10^1$ | $5.48 \times 10^1$ | $4.78 \times 10^1$ | $4.90 \times 10^1$ | $6.09 \times 10^1$ |
| Adaptation | same as above | same as above | 0 | $5.58 \times 10^{-3}$ | $5.53 \times 10^{-3}$ |
| Inference* | $4.11 \times 10^{-1}$ | $5.44 \times 10^{-1}$ | $4.19 \times 10^{-1}$ | $4.61 \times 10^{-1}$ | $4.35 \times 10^{-1}$ |

*Table 8.* Comparison of hypervolume differences (mean and quartiles). $Q_1$, $Q_2$, and $Q_3$ denote the first, second, and third quartiles, respectively. All values denote differences from GFB (GFB − competitor); positive values indicate that the proposed method performs better.

| Dataset | Method | Mean | $Q_1$ | $Q_2$ | $Q_3$ |
|---|---|---|---|---|---|
| CelebA | vs. EPO | 0.0465 | 0.0692 | 0.0695 | 0.0808 |
|  | vs. FairBiNN | 0.1279 | 0.0891 | 0.1100 | 0.1759 |
|  | vs. FairBayes | 0.0941 | 0.0874 | 0.0927 | 0.0976 |
|  | vs. YOTO | 0.2130 | 0.1087 | 0.2060 | 0.2877 |
| UTKFace | vs. EPO | 0.0125 | -0.0316 | 0.0321 | 0.0513 |
|  | vs. FairBiNN | -0.0295 | -0.0428 | -0.0313 | -0.0131 |
|  | vs. FairBayes | 0.0379 | 0.0184 | 0.0259 | 0.0673 |
|  | vs. YOTO | 0.1100 | 0.0545 | 0.1415 | 0.1632 |
| Adult | vs. EPO | 0.0770 | 0.0481 | 0.0829 | 0.0844 |
|  | vs. FairBiNN | 0.0452 | 0.0170 | 0.0426 | 0.0564 |
|  | vs. FairBayes | 0.0157 | 0.0203 | 0.0244 | 0.0306 |
|  | vs. YOTO | 0.1515 | 0.1247 | 0.1443 | 0.1671 |
| COMPAS | vs. EPO | -0.0027 | -0.0408 | -0.0243 | 0.0552 |
|  | vs. FairBiNN | -0.0238 | -0.0331 | -0.0284 | 0.0047 |
|  | vs. FairBayes | 0.0566 | -0.0151 | 0.0019 | 0.1459 |
|  | vs. YOTO | 0.1722 | 0.1260 | 0.1854 | 0.2894 |

### E.5.2. QUARTILE ANALYSIS OF HV AND INVERTED HV DIFFERENCES

To assess the reliability of the results across random seeds, we report the mean and quartiles of the seed-wise differences for both HV and inverted HV in Tables 8 and 9, respectively. All values are computed as GFB minus the corresponding competitor. For HV, positive values indicate that GFB achieves higher HV, whereas negative values indicate that the competitor achieves higher HV. When all quartiles are positive, GFB consistently outperforms the competitor across seeds; when all quartiles are negative, the competitor consistently achieves higher HV. Mixed signs among the quartiles indicate that the relative performance varies across seeds. For inverted HV, negative values indicate that GFB achieves lower inverted HV. When all quartiles are negative, GFB consistently has fewer poorly performing dominated solutions than the competitor across seeds.

Compared with EPO, all quartiles are positive on CelebA and Adult, indicating that GFB consistently achieves higher HV across seeds on these datasets. On UTKFace, only the first quartile is negative, while the median and third quartile are positive. This suggests that GFB generally outperforms EPO, although the advantage is not uniform across all seeds. On COMPAS, only the third quartile is positive. This indicates that EPO achieves higher HV in typical runs, although GFB outperforms EPO in some seeds. However, the inverted HV comparison in Table 9 provides a complementary view: all quartiles are negative on COMPAS, meaning that GFB consistently achieves smaller inverted HV than EPO across seeds. Thus, although EPO has an advantage in terms of standard HV on COMPAS, GFB avoids poorly performing trade-off points more consistently. Overall, these results suggest that GFB is competitive with, and often outperforms, EPO, while additionally providing post-hoc controllability.

We next compare GFB with FairBiNN. On CelebA and Adult, all quartiles are positive, showing that GFB consistently achieves higher HV across seeds. On COMPAS, only the third quartile is positive, indicating that FairBiNN achieves higher

*Table 9.* Comparison of inverted Hypervolume differences (mean and quartiles). $Q_1$, $Q_2$, and $Q_3$ denote the first, second, and third quartiles, respectively. All values denote differences from GFB (GFB − competitor); positive values indicate that the proposed method performs better.

| Dataset | Method | Mean | $Q_1$ | $Q_2$ | $Q_3$ |
|---|---|---|---|---|---|
| CelebA | vs. EPO | -0.2295 | -0.3732 | -0.1988 | -0.1032 |
| | vs. FairBiNN | -0.1561 | -0.1569 | -0.1466 | -0.1375 |
| | vs. FairBayes | -0.0354 | -0.0461 | -0.0300 | -0.0218 |
| | vs. YOTO | -0.0955 | -0.0620 | 0.0139 | 0.0150 |
| UTKFace | vs. EPO | -0.3432 | -0.7351 | -0.0696 | -0.0606 |
| | vs. FairBiNN | -0.1569 | -0.1557 | -0.1443 | -0.1169 |
| | vs. FairBayes | -0.0546 | -0.0766 | -0.0556 | -0.0553 |
| | vs. YOTO | -0.0548 | -0.0976 | -0.0673 | -0.0456 |
| Adult | vs. EPO | -0.2287 | -0.2617 | -0.2257 | -0.1831 |
| | vs. FairBiNN | -0.0993 | -0.1145 | -0.0976 | -0.0798 |
| | vs. FairBayes | -0.0015 | -0.0103 | -0.0101 | 0.0090 |
| | vs. YOTO | -0.1800 | -0.1630 | -0.1619 | -0.1515 |
| COMPAS | vs. EPO | -0.0809 | -0.1011 | -0.0714 | -0.0678 |
| | vs. FairBiNN | -0.0756 | -0.0798 | -0.0756 | -0.0645 |
| | vs. FairBayes | -0.0326 | -0.0464 | -0.0419 | -0.0383 |
| | vs. YOTO | -0.2263 | -0.2937 | -0.2009 | -0.1565 |

HV in typical runs, while GFB outperforms FairBiNN in some seeds. On UTKFace, all quartiles are negative, showing that FairBiNN consistently achieves higher HV across seeds. Overall, GFB achieves higher HV than FairBiNN on CelebA and Adult, remains competitive on COMPAS, and underperforms on UTKFace. Nevertheless, The inverted HV comparison provides a complementary view. As shown in Table 9, all quartiles of the inverted HV differences are negative across all datasets, indicating that GFB consistently achieves smaller inverted HV than FairBiNN across seeds. This suggests that, although FairBiNN shows an advantage under the standard HV metric on COMPAS and UTKFace, GFB more consistently avoids poorly performing dominated solutions. These results suggest that GFB provides a more stable trade-off curve overall, while additionally offering post-hoc controllability, which FairBiNN does not provide.

We then compare GFB with YOTO. All quartiles of the HV differences are positive across all datasets. This indicates that GFB consistently achieves higher HV than YOTO across seeds. Moreover, the inverted HV differences are negative for all datasets, showing that GFB also avoids poorly performing dominated solutions more consistently. Thus, GFB clearly outperforms YOTO in terms of both HV and inverted HV.

Finally, we compare GFB with FairBayes. All quartiles are positive on CelebA, UTKFace, and Adult, indicating that GFB consistently achieves higher HV across seeds on these datasets. On COMPAS, only the first quartile is negative, while the median and third quartile are positive. This suggests that GFB generally achieves higher HV than FairBayes, although the advantage is not uniform across all seeds. Nevertheless, GFB achieves a higher mean HV with a smaller standard deviation, suggesting more favorable and stable performance overall on COMPAS. The inverted HV comparison provides further evidence for this interpretation. Against FairBayes, all quartiles of the inverted HV differences are negative across all datasets, indicating that GFB more consistently avoids poorly performing trade-off points.

We also examine the standard deviations reported in Table 1 and Table 2 to assess variability across random seeds. For HV, GFB shows small standard deviations on most datasets; in particular, it achieves the smallest standard deviation on CelebA, Adult, and COMPAS. A similar trend is observed for inverted HV, where GFB again shows small standard deviations, including the smallest value on CelebA and COMPAS. These results suggest that the proposed boundary-concentration mechanism improves the fairness-accuracy trade-off over post-hoc controllable methods without introducing additional variability across random seeds.

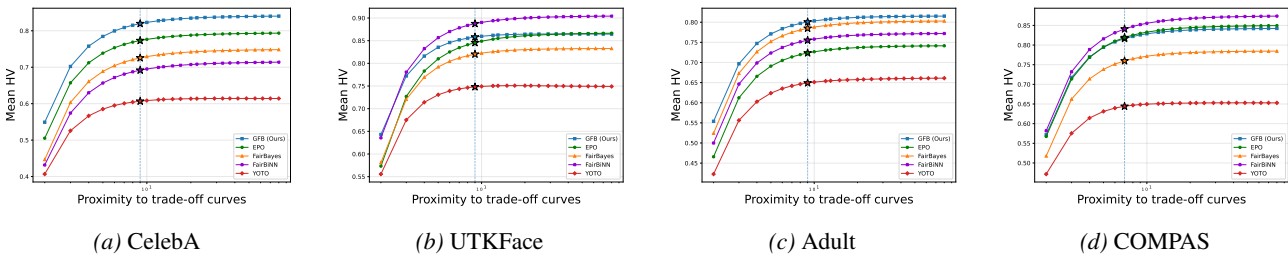

*Figure 5.* Sensitivity of HV to the reference-point choice. The horizontal axis represents the proximity of the reference point to the trade-off curves, represented by $N - 1$, and the vertical axis represents the HV value. The star marks the value used for the main HV results in Table 1.

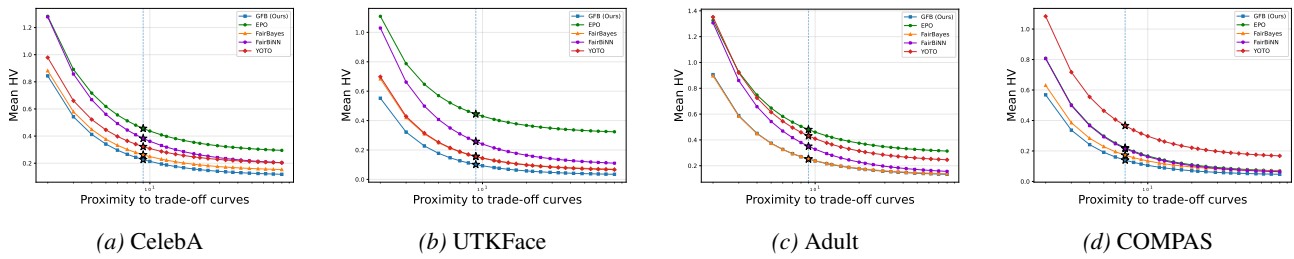

*Figure 6.* Sensitivity of inverted HV to the reference-point choice. The horizontal axis represents the proximity of the reference point to the trade-off curves, represented by $N - 1$, and the vertical axis represents the inverted HV value. The star marks the value used for the main inverted HV results in Table 2.

### E.5.3. SENSITIVITY ANALYSIS OF HV AND INVERTED HV

We further analyze the sensitivity of both HV and inverted HV rankings to the choice of the reference point. Specifically, we vary the distance between the reference point and the trade-off curves and examine how the rankings change. To do so, we vary the value of $N - 1$ in the reference-point construction in Equation (27) over logarithmically spaced values from 2 to 80.

As shown in Figure 5, the HV rankings remain unchanged on CelebA and Adult, and only limited changes are observed on UTKFace and COMPAS. We observe a similar trend for the inverted HV: although slight ranking changes appear on COMPAS and some curves overlap on datasets other than CelebA, the rankings do not change substantially across the examined range of reference points. These results suggest that the main conclusions drawn from HV and inverted HV are not strongly affected by the choice of the reference point.

