# OpenReview forum: "Fair Classification with Efficient and Post-hoc Controllable Fairness-Accuracy Trade-off"
_ICML.cc/2026/Conference — ICML 2026 spotlight_

### Official Review · Reviewer_h3Mx · 2026-02-22

**Soundness:** 3
**Presentation:** 3
**Significance:** 3
**Originality:** 3
**Overall Recommendation:** 4
**Confidence:** 3

**Summary:**

The paper proposes a novel framework, Guidance to Fairest-Boundary (GFB), to achieve efficient and post-hoc controllable fairness-accuracy trade-offs. GFB uses a bi-level optimization approach (specifically adapting MA-SOBA) to learn feature representations that are explicitly optimized for subsequent post-processing methods (like FairBayes). By concentrating features near the decision boundary of the fairest classifier, GFB allows for flexible adjustment of fairness constraints at inference time without the need for retraining, while maintaining high trade-off efficiency.

**Compliance With Llm Reviewing Policy:**

Affirmed.

**Final Justification:**

This paper makes a respectable technical contribution to post-hoc controllable fairness via bi-level optimization, with strong empirical results across image and tabular data. The rebuttal partially addressed my concerns: the max-margin interpretation is theoretically plausible but lacks empirical validation under distribution shifts; the 1.68× training overhead should be explicitly acknowledged in the efficiency claims; and while the EOp extension is welcome, the inability to handle EO and intersectional fairness remains a practical limitation.

Given the methodological novelty, practical relevance for deployment with shifting fairness requirements, and the authors' commitment to address presentation issues, I maintain my recommendation of **Weak Accept**.

**Key Questions For Authors:**

1. Could you comment on the risk of overfitting associated with concentrating features near the decision boundary? Have you performed any experiments involving distribution shifts or robustness checks to ensure the learned representations remain valid for test data that might drift slightly from the training distribution?

2. How does the training time of GFB (using MA-SOBA) compare to YOTO and standard ERM?

3. How difficult would it be to extend the proposed bi-level objective to other fairness metrics like Equalized Odds? Is the restriction to DP a fundamental limitation of the current theoretical derivation?

**Limitations:**

The authors acknowledge limitations regarding binary attributes and Demographic Parity (not adequately discussed). However, the discussion may be expanded to include:

1.  **Generalization Risks**: The strategy of concentrating features near the decision boundary contradicts max-margin principles and may hurt robustness.
2.  **Training Overhead**: The computational cost of the bi-level optimization compared to standard training.
3.  **Scalability**: Whether the method truly scales to the Large Language Models used as motivation in the introduction.

**Strengths And Weaknesses:**

### Strengths

**1. Practical Significance**: The problem of post-hoc controllability is highly relevant for real-world deployment, where fairness requirements may shift (e.g., due to policy changes) after a model is deployed. The proposed method addresses the trade-off efficiency issue inherent in traditional post-processing methods.

**2. Methodological Novelty**: The idea of coupling representation learning via bi-level optimization with post-processing is an interesting direction. It attempts to bridge the gap between the flexibility of post-processing and the performance of in-processing.

**3. Experimental Coverage**: The paper evaluates the method on both image (CelebA, UTKFace) and tabular datasets (Adult, COMPAS), demonstrating its applicability across different data modalities.

### Weaknesses

**1. Concern on Generalization (Boundary Concentration)**: The paper argues that concentrating features near the decision boundary improves trade-off efficiency. While this might mathematically allow for sensitive threshold adjustments on the training set, it seems to contradict the standard "Max-Margin" principle in machine learning, which suggests that features should be well-separated from the boundary for better robustness and generalization. I am concerned that this "boundary concentration" strategy might lead to poor generalization on unseen data, especially under slight distribution shifts.

**2. Limited Metrics**: The paper exclusively evaluates Demographic Parity (DP) for fairness and Accuracy for utility.

**3. Missing Experimental Details**:

- **Table 2**: The caption of Table 2 states that "Parenthesized values denote the standard deviation," but the table itself only contains mean values.
- **Training Efficiency**: While the method boasts *inference* efficiency, it uses a bi-level optimization process for training. The paper lacks a comparison of *training* time and computational cost against baselines like YOTO or standard ERM. If training is prohibitively slow, the "efficiency" argument is partially weakened.

**4. Scope Limitations**: The current formulation and evaluation are limited to binary classification with binary sensitive attributes. While the authors briefly mention this in the conclusion, a more detailed discussion on the challenges of extending this to multi-class or multi-attribute settings would be beneficial.

**5. Missing Impact Statement**: The submission misses a dedicated **Impact Statement** section, which is a mandatory requirement for ICML 2026 submissions.

**6. The code is not available**.

---

> ### Author Rebuttal · Authors · 2026-03-31
>
> Thank you for your positive assessment and valuable feedback. We will answer your questions below.
>
> ---
> ### Q1. Consistency with the Max-margin Principle
> We would like to clarify that GFB is consistent with the max-margin principle. We interpret the margin as the distance from a feature to the accuracy-maximizing decision boundary. GFB concentrates features near the fairness-maximizing decision boundary, which is typically separated distant from the accuracy-maximizing boundary. Therefore, these features have a large margin. Moreover, this concentration is applied only to features that lie near the accuracy-maximizing boundary against the fairness-maximizing boundary. Moving these features toward the fairness-maximizing boundary only means to enlarge their margin.
>
>
> ---
> ### Q2. Training time of GFB compared to other methods
> We evaluated computational time on the CelebA dataset, as shown in Table 5. In Table 5, “Train” reports the training time for 100 epochs, excluding hyperparameter tuning, “Adaptation” denotes the additional computation time required when changing the trade-off parameter (threshold estimation for FairBayes/GFB, retraining for EPO), and “Inference” denotes the time required to produce predictions on the test set. For FairBayes and GFB, the training time includes one forward pass on a holdout set used for threshold estimation to compute logits.
>
> **Table 5: Wall-clock time (seconds) on the CelebA dataset for all methods (averaged over 5 seeds). * per parameter for EPO/YOTO, 10 parameters for FairBayes/GFB.**
> | |EPO|YOTO|FairBayes|GFB(Ours)|
> |-|-|-|-|-|
> |Train|$4.58\times10^3$|$1.77\times10^3$|$1.78\times10^3$|$2.98\times10^3$|
> |Adaptation|same as above|0|$5.10\times10^{-2}$|$5.16\times10^{-2}$|
> |Inference*|5.72|5.03|5.52|5.55|
>
> GFB requires approximately 1.68 times the training time of YOTO and FairBayes. However, we believe that this increase in training time does not undermine the overall benefits of our method, as it achieves improvements in HV of about 0.22 and 0.10 points on CelebA compared to YOTO and FairBayes, respectively, as shown in Table 1 provided in our response to Reviewer NnkY (Q1).
>
> ---
> ### Q3. Limited fairness metrics
> We agree that the current framework is tailored to DP, and its scope is therefore limited. However, our additional analysis motivated by your comment found that our framework can be extended to Equality of Opportunity (EOp).
>
> Specifically, for EOp, Zeng et al. (2024) show that the fair Bayes-optimal classifier can be expressed as a thresholding rule whose threshold varies monotonically with the fairness tolerance $\delta$. Based on this structure, by defining
> $$T_a(X') = p_{a,1}\left(2 - \frac{1}{\eta_a(X')}\right),$$
> where $p_{a,1}=P(A=a, Y=1)$, the conditions of Theorem 4.1 are satisfied, and the same decomposition result follows. Moreover, for Theorem 4.2, by adapting the DDP-specific surrogate function in Eq. (15) to a form corresponding to EOp, the implicit gradient can be derived in the same manner.
>
> These observations suggest that our theoretical framework can be naturally extended to EOp, thereby mitigating the limitation in scope. We will include a detailed derivation of this extension in the appendix of the revised manuscript.
>
> For Equalized Odds (EO), the fair Bayes-optimal classifier does not admit a representation of the form $I(T_a(X') > (2a-1)t)$, and thus our analysis does not directly apply. Extending it to EO requires additional technical development.
>
> ---
> ### Missing Impact Statement
> Thank you for pointing out the absence of an Impact Statement in the submitted version. In the final version, we will include the following statement: "This paper presents work whose goal is to advance the field of machine learning. There are many potential societal consequences of our work, none of which we feel must be specifically highlighted here."
>
> ---
> ### Code availability
> We will make a cleaned and well-documented version of the code publicly available upon publication.
>
> ---
> ## References
> Zeng, X., Cheng, G., & Dobriban, E. (2024). Bayes-Optimal Fair Classification with Linear Disparity Constraints via Pre-, In-, and Post-processing. ArXiv, abs/2402.02817.

---

> > ### Author Rebuttal · Reviewer_h3Mx · 2026-04-01
> >
> > Thank you for the clarifications. While the additional results address some concerns, I remain partially unresolved on the following points:
> >
> > Q1: The argument that concentrating features near the fairness boundary "enlarges the margin" relies on a specific interpretation. Without empirical validation under distribution shifts or noise, the concern that boundary concentration may hurt robustness remains.
> >
> > Q2: Table 5 shows 1.68× training time overhead vs. FairBayes/YOTO. Given the title emphasizes "efficient" trade-offs, the paper should explicitly clarify whether "efficiency" refers only to inference, and discuss whether this training cost is acceptable for practitioners.
> >
> > Q3: While the extension to EOp is well noted, the inability to handle Equalized Odds (EO) and intersectional fairness remains a significant limitation for deployment. Please further discuss this restriction in the limitations section.

---

> > > ### Author Response · Authors · 2026-04-06
> > >
> > > We thank you for your careful reading and thoughtful understanding of our response. Below, we address your questions.
> > >
> > > ---
> > >
> > > ### Q1. Robustness under distribution shifts and noise
> > > We agree that robustness under distribution shifts is an important consideration. However, such settings are outside the scope of the current work, and we would like to emphasize that the contributions of our paper remain valid regardless of robustness under distribution shift.
> > >
> > > Regarding robustness to noise, we note that our experiments provide some evidence of stable performance across seeds. Table 1 in our response to Reviewer NnkY (Q1) reports the mean and standard deviation of HV, showing that the proposed method maintains relatively small standard deviations across datasets. In addition, the table below reports the interquartile range (IQR) of HV for each method. The results show that the proposed method does not exhibit unusually large IQR compared to other methods. Taken together, these results suggest that the proposed method exhibits stable behavior with respect to training stochasticity and does not introduce additional sensitivity to noise compared to existing methods.
> > >
> > > **Table** : Comparison of Interquartile Range (IQR) of HV
> > >
> > > | Dataset | EPO | YOTO | FairBayes | GFB (Ours) |
> > > |--------|-----|------|-----------|-----|
> > > | CelebA | 0.0073 | 0.1913 | 0.0292 | 0.0026 |
> > > | UTKFace | 0.0397 | 0.0643 | 0.0160 | 0.0409 |
> > > | Adult | 0.1008 | 0.0435 | 0.0672 | 0.0393 |
> > > | COMPAS | 0.0104 | 0.1839 | 0.1312 | 0.0249 |
> > >
> > > ---
> > >
> > > ### Q2. Interpretation of efficiency
> > > We emphasize that “efficiency” in our paper refers primarily to the ability to adjust the fairness–accuracy trade-off **at inference time**. We recognize that this was not sufficiently clear and may have been misleading. We will revise the introduction to make this scope explicit.
> > >
> > > ---
> > >
> > > ### Q3. Limitations regarding EO and intersectional fairness
> > > We agree that the discussion of practical limitations was insufficient. In the final version, we will expand the limitations section to explicitly discuss these restriction, and clarify the challenges in extending the framework to EO and intersectional fairness.

---

### Official Review · Reviewer_NnkY · 2026-02-27

**Soundness:** 3
**Presentation:** 4
**Significance:** 3
**Originality:** 3
**Overall Recommendation:** 5
**Confidence:** 4

**Summary:**

This paper proposes GFB (Guidance to Fairest-Boundary), a representation learning method that improves the trade-off efficiency of post-processing fair classifiers. The core idea is to learn feature representations that concentrate near the decision boundary of the most fair classifier, motivated by Theorem 4.1, which shows that accuracy gains from relaxing fairness constraints scale with the mass of the feature distribution near that boundary. The resulting training objective is formulated as a bi-level optimization problem and solved using MA-SOBA. At inference, the standard FairBayes post-processing step is applied, preserving post-hoc controllability. The method is evaluated on CelebA, UTKFace, Adult, and COMPAS against EPO, YOTO, and FairBayes using hypervolume as the primary metric.

**Compliance With Llm Reviewing Policy:**

Affirmed.

**Final Justification:**

GFB makes a clean and well-motivated contribution: the decomposition in Theorem 4.1 is principled, the bi-level training objective follows directly from it, and the post-hoc controllability framing is practically relevant. The rebuttal addressed my main concerns -- corrected variance tables with an acknowledged reference point error, a training cost comparison that now quantifies the post-hoc controllability advantage, and a sketch of an Equal Opportunity extension. The DDP justification for COMPAS remains thin, but this is a minor concern. The corrected results and computational comparison should appear in the final version.

**Key Questions For Authors:**

From above weaknesses:
1. Can standard deviations be added to all cells in Table 2, as indicated by the caption?
2. Why is demographic parity used for COMPAS rather than equalized odds? Does the GFB design principle extend to other fairness metrics?
3. Can training wall-clock time be reported for all methods on at least one dataset?
4. How sensitive are the HV rankings to the choice of reference point, particularly for COMPAS where the GFB vs. EPO margin is essentially zero?

**Limitations:**

The authors discuss limitations in Section 6: demographic parity restriction, binary sensitive attribute and label restriction, and directions for extension. This is honest and appropriately scoped. One gap is the absence of any discussion of fairness metric choice per dataset, which bears directly on the validity of the experimental comparisons.

**Strengths And Weaknesses:**

Strengths:
1. The problem framing is clear and well-motivated. Post-processing methods are post-hoc controllable but efficiency-limited; in-processing methods are efficient but not post-hoc controllable. This is a genuine and practically relevant challenge, and the paper positions GFB cleanly relative to both categories. The LLM regulatory compliance example in Section 1.1 is a concrete and appropriate motivating case.

2. The theory-to-method connection in Section 4 is the strongest part of the paper. Theorem 4.1 provides a principled design criterion. It shows that concentrating feature mass near the fairest boundary increases Accdep, which directly motivates the learning objective Ldist. The implicit gradient result in Theorem 4.2 addresses a genuine technical challenge in differentiating through the threshold search, and the proof in Appendix D is complete.

3. The hypervolume metric is appropriate for comparing Pareto front efficiency across methods. The evaluation covers both image and tabular settings with two post-hoc controllable baselines (FairBayes, YOTO) and one in-processing baseline (EPO). GFB achieves the highest HV across all four datasets, a consistent result.

Weaknesses:
1. The evaluation reports averages over 5 runs, and the Table 2 caption states that parenthesized values denote standard deviations, but no parenthesized values are present in the rendered table. Furthermore, std statistics derived from 5 runs is dubious. The significance of GFB's improvements is therefore unverifiable from the paper as submitted. On Adult, the HV delta over FairBayes is 0.8716 vs. 0.8633, and on COMPAS over EPO it is 0.9110 vs. 0.9109. These deltas are small enough that standard deviations are essential to interpret. The authors should report variance for all entries in Table 2 and discuss whether the observed improvements are statistically meaningful.

2. The method is restricted to difference of demographic parity (DDP) with binary sensitive attributes and binary labels. This is acknowledged in Section 6, but the practical implication is not discussed adequately in the experimental section. COMPAS is a dataset where equalized odds is arguably more appropriate than demographic parity given the prediction context, yet the paper uses DDP without justification. A discussion of why DDP is the right metric for each dataset, and whether the GFB design principle extends to equalized odds or equal opportunity, would substantially strengthen the paper. Extending the theoretical analysis to at least one additional fairness metric, and providing a clear argument for why the current result does not generalize, would address this.

3. The computational cost of GFB's training procedure is not reported. The paper's motivation emphasizes avoiding retraining costs, and YOTO's inference overhead is discussed explicitly in Appendix A. However, GFB's bi-level MA-SOBA training cost relative to standard ERM or YOTO training is never quantified. For the post-hoc controllability argument to be complete, the training cost comparison should be included alongside inference cost. Reporting wall-clock training time for all methods across at least one dataset would clarify this.

4. The hypervolume metric is sensitive to the choice of reference point, and the sensitivity of the reported rankings to this choice is not analyzed. On COMPAS, GFB (0.9110) barely outperforms EPO (0.9109). Small perturbations to the reference point could plausibly change this ranking. The reference point procedure follows Ishibuchi et al. (2018) and is principled, but a brief sensitivity analysis varying the reference point, or reporting trade-off curves alongside HV for all datasets rather than only Figure 3, would make the comparison more robust.

------
------

Summary:

The theory is clean and the method-motivation connection is direct, which are genuine strengths. The paper falls just short because the variance is missing from the primary results table, the computational cost comparison is incomplete, and the single-fairness-metric scope is not adequately justified for the datasets used. These are all addressable, and the core contribution would support acceptance with revisions. I would encourage the authors to add variance to Table 2, add a training cost comparison, and discuss the DDP metric choice per dataset before resubmission.

---

> ### Author Rebuttal · Authors · 2026-03-31
>
> Thank you for your careful reading and valuable feedback. We would like to address your questions below.
>
> ---
> ### Q1. Statistical significance of the proposed method / Q4. Sensitivity of HV rankings to the choice of reference point
> We acknowledge that our results are not conclusive enough to claim clear statistical superiority. Nevertheless, we contend that our method provides significant practical advantages, including superior stability and the critical advantage of post-hoc controllability.
>
> To support this, we report the updated statistics in Table 1 and Table 2. Table 1 shows the mean and standard deviation of HV, while Table 2 reports the mean and quartiles of HV differences across seeds. In this process, we identified an error in the computation of the reference point for HV in the original paper, for which we apologize. The reference point was computed from all solutions; here, following Dushatskiy et al. (2023) and Ishibuchi et al. (2018), we report the corrected results based on the Pareto front of each trial. Importantly, this issue was limited to the reference point computation and does not affect the relative positions of the trade-off curves. Figure 3 shows that our method occupies the high-accuracy, low-unfairness region of the trade-off curve, regardless of the choice of reference point or HV.
>
> Comparing our method with EPO, our method outperforms it on CelebA and Adult, and shows competitive performance on UTKFace. On COMPAS, our method performs slightly worse. We emphasize that this still supports the practical advantage of our approach: despite offering post-hoc controllability, our method achieves the competitive performance to EPO, which lacks this property.
>
> Compared with FairBayes, our method achieves a positive first quartile ($Q_1$) on CelebA, UTKFace, and Adult, indicating that it outperforms FairBayes in at least 75% of the runs. On COMPAS, while FairBayes attains a higher median ($Q_2$), our method achieves a higher mean HV (0.8479 vs. 0.7901) and a substantially smaller standard deviation (±0.030 vs. ±0.088), demonstrating more stable performance. Overall, our method shows competitive or improved performance across settings, with notably higher stability across runs.
>
>
> **Table 1**: Comparison of Hypervolume (Mean ± Std)
> |Dataset|EPO|YOTO|FairBayes|GFB (Ours)|
> |-|-|-|-|-|
> |CelebA |0.8178 ± 0.058|0.6410 ± 0.150|0.7702 ± 0.030|**0.8652 ± 0.005**|
> |UTKFace|0.8572 ± 0.023|0.7448 ± 0.083|0.8256 ± 0.022|**0.8587 ± 0.027**|
> |Adult  |0.7528 ± 0.073|0.6722 ± 0.044|0.8146 ± 0.037|**0.8271 ± 0.036**|
> |COMPAS |**0.8542 ± 0.060**|0.6611 ± 0.156|0.7901 ± 0.088|0.8479 ± 0.030|
>
>
> **Table 2**: Comparison of Hypervolume differences (Mean and Quartiles). $Q_1$, $Q_2$, and $Q_3$ denote the first, second, and third quartiles, respectively. All values denote differences from GFB (GFB−competitor); positive values indicate that the proposed method performs better.
> |Dataset|Method|Mean|$Q_1$|$Q_2$|$Q_3$|
> |-|-|-|-|-|-|
> |**CelebA**|vsEPO|0.0474|-0.0633|0.0678|0.0698|
> ||vsFairBayes|0.0950|0.0859|0.0902|0.1021|
> ||vsYOTO|0.2241|0.1171|0.2232|0.3149|
> |
> |**UTKFace**|vsEPO|0.0015|-0.0394|0.0157|0.0438|
> ||vsFairBayes|0.0332|0.0114|0.0134|0.0636|
> ||vsYOTO|0.1139|0.0552|0.1467|0.1784|
> |
> |**Adult**|vsEPO|0.0743|0.0490|0.0733|0.0863|
> ||vsFairBayes|0.0125|0.0166|0.0248|0.0399|
> ||vsYOTO|0.1549|0.1363|0.1412|0.1659|
> |
> |**COMPAS**|vsEPO|-0.0064|-0.0412|-0.0229|0.0448|
> ||vsFairBayes|0.0578|-0.0142|-0.0035|0.1607|
> ||vsYOTO|0.1868|0.1211|0.1830|0.3298|
>
>
> ---
> ### Q2. Justification for using DDP and generalizability to other fairness metrics
>
> We note that our experiments are designed to evaluate the performances of both the existing methods and our methods in settings where demographic parity is desired. The COMPAS dataset is a widely used benchmark for assessing fairness even in terms of demographic parity in prior work. Therefore, we believe that evaluating performance under demographic parity on COMPAS dataset remains informative.
>
> For discussion on the extensibility to other fairness metrics, please refer to our response to Reviewer h3Mx (Q3).
>
> ---
> ### Q3. The computational cost of training procedure
> Please refer to our response to Reviewer h3Mx (Q2) for detailed timing results.
>
> While GFB requires more training time than YOTO and FairBayes, it delivers more stable and improved performance across most datasets. We therefore consider this additional training cost to be justified by the overall performance gains.
>
> ---
> ## References
> Arkadiy Dushatskiy, Alexander Chebykin, Tanja Alderliesten, and Peter A.N. Bosman. 2023. Multi-objective population based training. In Proceedings of the 40th International Conference on Machine Learning (ICML'23), Vol. 202. JMLR.org, Article 359, 8969–8989.

---

> > ### Author Rebuttal · Reviewer_NnkY · 2026-03-31
> >
> > Thank you for the detailed response. The corrected variance tables and quartile breakdowns are useful and partially address W1 and W4. I'm revising my score to weak accept.
> >
> > W1/W4 (variance and reference point): The reference point error changes the headline claim -- GFB is not #1 on all four datasets with the corrected numbers. That said, on the datasets where GFB leads, the margins are wider than the deficit on COMPAS, and the stability advantage there (std 0.030 vs. 0.060 for EPO) is real. The trade-off curves in Figure 3 support this reading independently of HV. This is a weaker claim than the submission makes, but not one that undermines the contribution.
> >
> > W2 (DDP for COMPAS): "Prior work also uses DDP on COMPAS" is not a justification. COMPAS predicts two-year recidivism; the well-known concern in this setting is disparate false positive rates across racial groups, which DDP does not capture. The authors ought to either argue that DDP is appropriate here on the merits or acknowledge that the metric choice was driven by the current method's scope rather than the dataset's context.
> >
> > W3 (training cost): The response acknowledges GFB takes more training time than YOTO and FairBayes but gives no numbers. The post-hoc controllability framing requires this comparison to be quantified.
> >
> > W2 and W3 remain unaddressed, and the corrected Table 2 should appear in the final version. These are fixable in revision, and the core contribution is sound enough to warrant acceptance conditional on those changes.

---

> > > ### Author Response · Authors · 2026-04-06
> > >
> > > We sincerely thank you for your thoughtful reconsideration and constructive feedback. Below, we provide additional responses to your comments.
> > >
> > > ---
> > >
> > > ### W2. Appropriateness of DDP for COMPAS
> > > We would like to clarify that we do not claim demographic parity is the most appropriate fairness criterion for the COMPAS dataset. Rather, our position is that there is no single universally appropriate fairness definition for a given dataset.
> > >
> > > We were aware that the original analysis by ProPublica focused on fairness in terms of equalized odds. However, this only reflects the particular perspective and priorities of that analysis, rather than establishing equalized odds as the definitive or universally appropriate fairness criterion for COMPAS. Different communities or stakeholders, even when considering the same application context, may prioritize different notions of fairness depending on their values and concerns.
> > >
> > > From the perspective of advancing fair machine learning, it is important to account for this diversity of viewpoints. Restricting evaluation only to fairness definitions that are currently considered most appropriate for specific datasets may overlook these differing perspectives on fairness. Therefore, evaluating our method on the COMPAS dataset under demographic parity is meaningful.
> > >
> > > ---
> > >
> > > ### W3. Quantitative comparison of training time
> > > We apologize that, due to space limitations in our initial response, the quantitative results were provided only by referring to our response to Reviewer h3Mx (Q2). Below, we provide the wall-clock time for each method on the CelebA dataset again for clarity.
> > >
> > > “Train” reports the training time for 100 epochs, excluding hyperparameter tuning, “Adaptation” denotes the additional computation time required when changing the trade-off parameter (threshold estimation for FairBayes/GFB, retraining for EPO), and “Inference” denotes the time required to produce predictions on the test set. For FairBayes and GFB, the training time includes one forward pass on a holdout set used for threshold estimation to compute logits.
> > >
> > > **Table: Wall-clock time (seconds) on the CelebA dataset for all methods (averaged over 5 seeds). * per parameter for EPO/YOTO, 10 parameters for FairBayes/GFB.**
> > > | |EPO|YOTO|FairBayes|GFB(Ours)|
> > > |-|-|-|-|-|
> > > |Train|$4.58\times10^3$|$1.77\times10^3$|$1.78\times10^3$|$2.98\times10^3$|
> > > |Adaptation|same as above|0|$5.10\times10^{-2}$|$5.16\times10^{-2}$|
> > > |Inference*|5.72|5.03|5.52|5.55|

---

### Official Review · Reviewer_WMR7 · 2026-03-03

**Soundness:** 3
**Presentation:** 2
**Significance:** 2
**Originality:** 2
**Overall Recommendation:** 4
**Confidence:** 2

**Summary:**

This paper targets post-hoc controllable fair classification under demographic parity (DP): the ability to adjust the fairness–accuracy operating point after training without retraining. It builds on the FairBayes-style result that DP-constrained classifiers can be implemented by group-wise thresholding of a score/logit with a 1D parameter search at inference. The core contribution is a training-time approach that aims to improve the efficiency of this post-processing trade-off by learning representations that reshape the transformed score distribution to be favorable for post-hoc thresholding. The authors provide a characterization of how accuracy varies with the DP tolerance $\delta$, identifying a $\delta$-dependent term that becomes larger when transformed scores concentrate near the “most fair” decision boundary. Motivated by this, they propose Guidance to Fairest-Boundary (GFB): a representation-learning objective combining (i) a distance penalty that encourages logits in an interval to lie close to the fairest threshold, and (ii) a standard prediction loss, leading to a bi-level optimization setup solved with a gradient-based procedure. Experiments on CelebA, UTKFace, Adult, and COMPAS compare against in-processing and post-/hybrid baselines (e.g., EPO, YOTO, FairBayes) and report improved trade-off curves / hypervolume relative to post-hoc controllable baselines and competitive performance against an in-processing baseline without retraining.

**Compliance With Llm Reviewing Policy:**

Affirmed.

**Final Justification:**

My concerns are fully addressed. Solid work.

**Key Questions For Authors:**

n/a

**Limitations:**

yes.

**Strengths And Weaknesses:**

Soundness

strength:
- keep established FairBayes-style post-processing at inference while improving its frontier via representation learning; Theorem 4.1 provides a concrete motivation for the proposed concentrate mass near the fairest boundary principle.
- Method is specified in sufficient detail: explicit distance penalty and interval definition, combined with prediction loss; bilevel formulation and training procedure are described.
- Experimental evaluation spans multiple datasets and includes both in-processing and post-hybrid baselines; results include variability reporting.

weakness:
- The optimization is effectively heuristic for a bilevel problem; soundness would be strengthened by clearer discussion of convergence/stability assumptions, sensitivity to inner-loop/EMA choices, and failure cases when the implicit-gradient approximation is inaccurate.
- The theoretical insight is suggestive (a sufficient-style condition for better trade-offs) but may not fully capture all factors affecting learned representations; the paper could be clearer about when the theory is expected to predict practice.

Presentation

weakness:
- When introducing key parameters such as $t$ and $\tau$, it will help readers follow if the meaning of such parameters are also introduced.
- Novelty boundaries could be sharper: the paper should more clearly separate what is inherited from FairBayes at inference from what is new in training/representation learning to avoid confusion about what is being contributed.

Significance

strength:
- Post-hoc controllability is practically relevant (fairness requirements change, retraining is costly); improving post-processing efficiency could have deployment impact.
- Empirically, the method appears to improve trade-off curves/hypervolume over post-hoc controllable baselines and can be competitive with an in-processing baseline without retraining.

weakness:
- Scope appears specialized to DP and the threshold-style setting; significance would be stronger if the paper clarified extension to multi-group attributes or other constraints (e.g., equalized odds), or explicitly framed current limitations.

Originality

strength:
- Main novelty is the combination of (i) a structural characterization linking frontier efficiency to score mass near the fairest boundary and (ii) a representation-learning objective (GFB) designed to enforce that property while retaining post-hoc inference.

---

> ### Author Rebuttal · Authors · 2026-03-31
>
> Thank you for your careful reading and insightful comments. We address each of your concerns below.
>
> ---
>
> ## Soundness
> ### W1. Optimization convergence
> We would like to highlight that our method builds on MA-SOBA, whose convergence has been theoretically established. Specifically, MA-SOBA converges under the following assumptions:
> 1. First-order Lipschitz continuity of the outer objective and second-order Lipschitz continuity of the inner objective
> 2. Strong convexity of the inner objective
> 3. Boundedness of the gradient at the optimal solution of the inner problem
>
> Assumptions 1 and 3 can be satisfied by choosing a doubly differentiable $\psi$ with bounded first and second derivatives. In our experiments, we use the sigmoid function as $\psi$, which satisfies these conditions. Assumption 2 requires the ERM objective to be strongly convex with respect to the model parameters, which is generally not satisfied in practice. However, we note that local strong convexity around local minima may be enough for convergence, as gradient-based optimization typically remains within the local region. The local strong convexity can be encouraged in practice, e.g., by incorporating weight decay.
>
> ---
> ### w2. The scope of Theorem 4.1
> We would like to emphasize that Theorem 4.1 provides an exact decomposition of accuracy into a $\delta$-dependent component and a $\delta$-independent component. The $\delta$-dependent term is the key factor governing the trade-off efficiency and corresponds exactly to the area of the yellow region in Figure 2. Thus, the theoretical result is not merely a sufficient-style condition but offers a precise characterization of the trade-off. Motivated by this result, our proposed penalty term is designed to be proportional to the area of this yellow region.
>
> ---
> ## Presentation
> ### W1. Unclear separation between inherited components and novel contributions
> The key difference between our proposed method and FairBayes lies in the training procedure; we replace ERM with our proposed GFB algorithm, as illustrated in Figure 1. Our method retains the same prediction-time procedure as FairBayes. GFB introduces a novel penalty term into the ERM objective, encouraging feature representations to concentrate near the fairest boundary.
>
> ---
> ## Significance
> ### W1. Limited scope beyond DP and threshold-based settings
> We would like to note that threshold-style classifiers are not restrictive in the binary setting, as fair Bayes-optimal classifiers take a threshold form.
>
> We agree that extending our framework to more general settings, including non-binary targets and multiple sensitive attributes, is a crucial direction for future work. However, such extensions require substantial additional technical development. For example, as shown by Xian et al., the fair Bayes-optimal classifiers in these settings no longer admit simple thresholding structures. Consequently, extending our analysis to such cases is non-trivial and is left for future investigation.
>
> For extensions to other fairness notions, please refer to our response to Reviewer h3Mx (Q3).
>
> ---
> ## References
> Ruicheng Xian, Lang Yin, and Han Zhao. 2023. Fair and optimal classification via post-processing. In Proceedings of the 40th International Conference on Machine Learning (ICML'23), Vol. 202. JMLR.org, Article 1581, 37977–38012.

---

> > ### Author Rebuttal · Reviewer_WMR7 · 2026-04-02
> >
> > I appreciate authors response to my questions and concerns. My concerns are fully addressed. I will increase the score.

---

### Official Review · Reviewer_8ewj · 2026-03-12

**Soundness:** 4
**Presentation:** 2
**Significance:** 3
**Originality:** 3
**Overall Recommendation:** 4
**Confidence:** 4

**Summary:**

The paper studies the problem of training binary classifiers that remain robust to post-hoc fairness adjustments, focusing on demographic parity. The main insight is that different regions of the representation space retain predictive accuracy to different extents as the fairness threshold parameter varies. By optimizing the distribution of representations, it is therefore possible to learn models whose accuracy degrades more gracefully under fairness adjustments. Building on this observation, the authors propose a bilevel optimization algorithm that explicitly optimizes for robustness to post-hoc fairness constraints. The paper also addresses gradient computation challenges arising from the non-differentiability of the proposed model. Empirical results support the effectiveness of the proposed approach.

**Compliance With Llm Reviewing Policy:**

Affirmed.

**Key Questions For Authors:**

How would you compare with FairBiNN? Is there any good reason to omit it as a competitor?

Can you provide any evidence that the proposed approach is indeed more robust than competitors w.r.t. to post-hoc changing of the fairness threshold?
Note: I understand that not all competitors can be compared in this regards.

**Limitations:**

Yes

**Strengths And Weaknesses:**

## Strong points

- The problem addressed in the paper is interesting and timely.
- The paper provides strong theoretical support for the validity of the proposed approach.
- The empirical results are promising.

## Weak points

- The paper is quite notation-heavy. I would suggest simplifying the notation by dropping subscripts and other decorations when they are not strictly necessary.
- While the results are promising, the empirical evaluation is somewhat limited. The paper reports experiments on only four datasets and compares against three competitors. In this regard, FairBiNN [1] could be a strong additional baseline. One of the figures in [1] is very similar to those reported in this paper (to the point that the axes appear to have the same scale). Among the datasets considered in [1] there is Adult, and the method appears to obtain a competitive --if not better-- Pareto curve.
- The main selling point of the proposed approach is the possibility of changing the fairness threshold post-hoc without losing much accuracy. However, the paper does not include experiments that explicitly evaluate how resilient the method is in this respect. It would be particularly useful to contrast this property with competing methods.

Note: to the best of my understanding, the provided experiments are obtained by re-training all models from scratch. If instead there is some post-hoc change of the threshold, this is not clear from the paper. There is a comment saying that first the algorithm is compared with the post-hoc methods YOTO and FairBayes, but then everything is mixed up in figure 3 and table 2 and my interpretation is the one given above. Clearly I could be wrong about this.


### Minor points
- the reference point in Figure 3 is undefined.
- the caption of Table 2 says that "Parenthesized values denotes the standard deviation", but no parenthesized values are present.
- figure 1 is not much informative (in my opinion)

## References

[1] Yazdani-Jahromi, Mehdi, et al. "Fair bilevel neural network (FairBiNN): on balancing fairness and accuracy via Stackelberg equilibrium." Advances in Neural Information Processing Systems 37 (2024): 105780-105818.

---

> ### Author Rebuttal · Authors · 2026-03-31
>
> Thank you for your positive assessment and for the valuable feedback. We address your questions below.
>
> ---
>
> ### Q1. Comparison with FairBiNN
> We would like to emphasize that FairBiNN lacks post-hoc controllability; namely, it requires re-training whenever the trade-off parameter is changed. Even in cases where FairBiNN may achieve slightly better performance, this does not diminish the significance of our contribution.
>
> To further support this point, we compare our results with those reported in Figure 1(a) of the FairBiNN paper on the Adult dataset. As shown below, our method achieves competitive trade-offs.
>
> DDP level | FairBiNN Acc | GFB Acc | Diff
>  --- | --- | --- | ---
>  ≈0.16 | 0.8480 | 0.8531 | +0.0051
>  ≈0.10 | 0.8450 | 0.8474 | +0.0024
>  ≈0.05 | 0.8400 | 0.8397 | -0.0003
>  ≈0.00 | 0.8350 | 0.8332 | -0.0018
>
> ---
> ### Q2. Lack of explicit evaluation of post-hoc robustness
> We would like to clarify that the results shown for YOTO, FairBayes, and GFB in Figure 3 already reflect post-hoc performance. For those methods, a single model is trained once on the training dataset and then evaluated on the testing dataset under 10 different trade-off parameters in a post-hoc manner. Since EPO lacks the post-hoc controllability, each reported point for EPO corresponds to a separately trained model.
>
> We do, however, acknowledge that the goal of post-hoc controllability was not explicitly stated in the problem formulation. We will address this by revising Section 3.1 to clearly define post-hoc controllability as the ability to adapt to different trade-off parameters without retraining.

---

> > ### Author Rebuttal · Reviewer_8ewj · 2026-03-31
> >
> > The authors have adequately addressed my questions. Including the FairBiNN results in the paper would further strengthen it; however, since these results do not affect the overall assessment of the method, I offer this as a suggestion rather than a requirement.

---

### Decision · Program_Chairs · 2026-04-30

**Decision:**

Accept (spotlight)

**Comment:**

The paper contributes a method for post-hoc controllability of fairness-accuracy trade-offs without retraining. The theoretical component is a key strength, particularly the insight linking trade-off efficiency to feature concentration near the fairness boundary and the accompanying optimization framework. Empirically, the method shows promising results across multiple datasets and appears competitive with in-processing baselines while retaining post-hoc flexibility.

One main concern is that the method relies on a bilevel optimization procedure whose computational cost and stability are not well characterized, raising concerns about practical efficiency. Some discussion about computational cost is needed. Another concern is the limited discussion of generalization to other fairness notions such as equalized odds or more complex scenarios. Please include the discussion. Those concerns are relatively minor compared to the contribution of this work.


Overall, the paper presents an interesting idea with solid theoretical motivation. I lean toward accepting this work. In the final version, please also include the comparisons to the additional baselines (e.g., FairBiNN) and include the results on the statistical significance of the proposed method as well as the sensitivity of HV rankings to the choice of reference point.